# CAUSAL-AWARE GRAPH NEURAL ARCHITECTURE SEARCH UNDER DISTRIBUTION SHIFTS

## ABSTRACT

Graph neural architecture search (Graph NAS) has emerged as a promising approach for autonomously designing graph neural network architectures by leveraging the correlations between graphs and architectures. However, the existing methods fail to generalize under distribution shifts that are ubiquitous in real-world graph scenarios, mainly because the graph-architecture correlations they exploit might be *spurious* and varying across distributions. In this paper, we propose to handle the distribution shifts in the graph architecture search process by discovering and exploiting the *causal* relationship between graphs and architectures to search for the optimal architectures that can generalize under distribution shifts. The problem remains unexplored with the following critical challenges: 1) how to discover the causal graph-architecture relationship that has stable predictive abilities across distributions, 2) how to handle distribution shifts with the discovered causal graph-architecture relationship to search the generalized graph architectures. To address these challenges, we propose a novel approach, Causal-aware Graph Neural Architecture Search (**CARNAS**), which is able to capture the causal graph-architecture relationship during the architecture search process and discover the generalized graph architecture under distribution shifts. Specifically, we propose Disentangled Causal Subgraph Identification to capture the causal subgraphs that have stable prediction abilities across distributions. Then, we propose Graph Embedding Intervention to intervene on causal subgraphs within the latent space, ensuring that these subgraphs encapsulate essential features for prediction while excluding non-causal elements. Additionally, we propose Invariant Architecture Customization to reinforce the causal invariant nature of the causal subgraphs, which are utilized to tailor generalized graph architectures. Extensive experiments on synthetic and real-world datasets demonstrate that our proposed CARNAS achieves advanced out-of-distribution generalization ability by discovering the causal relationship between graphs and architectures during the search process.

## 1 INTRODUCTION

Graph neural architecture search (Graph NAS), aiming at automating the designs of GNN architectures for different graphs, has shown great success by exploiting the correlations between graphs and architectures. Present approaches (11; 25; 31) leverage a rich search space filled with GNN operations and employ strategies like reinforcement learning and continuous optimization algorithms to pinpoint an optimal architecture for specific datasets, aiming to decode the natural correlations between graph data and their ideal architectures. Based on the independently and identically distributed (I.I.D) assumption on training and testing data, existing methods assume the graph-architecture correlations are stable across graph distributions.

Nevertheless, distribution shifts are ubiquitous and inevitable in real-world graph scenarios, particularly evident in applications existing with numerous unforeseen and uncontrollable hidden factors like drug discovery, in which the availability of training data is limited, and the complex chemical properties of different molecules lead to varied interaction mechanisms (20). Consequently, GNN models developed for such purposes must be generalizable enough to handle the unavoidable variations in data distribution between training and testing sets, underlining the critical need for models that can adapt to and perform reliably under such varying conditions.

However, existing Graph NAS methods fail to generalize under distribution shifts, since they do not specifically consider the relationship between graphs and architectures, and may exploit the *spurious* correlations between graphs and architectures unintendedly, which vary with distribution shifts, during the search process. Relying on these spurious correlations, the search process identifies patterns that are valid only in the training data but do not generalize to unseen data. This results in good performance on the training distribution but poor performance when the underlying data distribution changes in the test set.

In this paper, we study the problem of graph neural architecture search under distribution shifts by capturing the *causal* relationship between graphs and architectures to search for the optimal graph architectures that can generalize under distribution shifts. The problem is highly non-trivial with the following challenges:

- How to discover the causal graph-architecture relationship that has stable predictive abilities across distributions?

- How to handle distribution shifts with the discovered causal graph-architecture relationship to search the generalized graph architectures?

To address these challenges, we propose the Causal-aware Graph NAS (**CARNAS**), which is able to capture the causal relationship, stable to distribution shifts, between graphs and architectures, and thus handle the distribution shifts in the graph architecture search process. Specifically, we design a *Disentangled Causal Subgraph Identification* module, which employs disentangled GNN layers to obtain node and edge representations, then further derive causal subgraphs based on the importance of each edge. This module enhances the generalization by deeply exploring graph features as well as latent information with disentangled GNNs, thereby enabling a more precise extraction of causal subgraphs, carriers of causally relevant information, for each graph instance. Following this, our *Graph Embedding Intervention* module employs another shared GNN to encode the derived causal subgraphs and non-causal subgraphs in the same latent space, where we perform interventions on causal subgraphs with non-causal subgraphs. Additionally, we ensure the causal subgraphs involve principal features by engaging the supervised classification loss of causal subgraphs into the training objective. We further introduce the *Invariant Architecture Customization* module, which addresses distribution shifts not only by constructing architectures for each graph with their causal subgraph but also by integrating a regularizer on simulated architectures corresponding to those intervention graphs, aiming to reinforce the causal invariant nature of causal subgraphs derived in module 1. We remark that the classification loss for causal subgraphs in module 2 and the regularizer on architectures for intervention graphs in module 3 help with ensuring the causality between causal subgraphs and the customized architecture for a graph instance. Moreover, by incorporating them into the training and search process, we make the Graph NAS model intrinsically interpretable to some degree. Empirical validation across both synthetic and real-world datasets underscores the remarkable out-of-distribution generalization capabilities of CARNAS over existing baselines. Detailed ablation studies further verify our designs. The contributions of this paper are summarized as follows:

- We are the first to study graph neural architecture search under distribution shifts from the causal perspective, by proposing the causal-aware graph neural architecture search (CARNAS), that integrates causal inference into graph neural architecture search, to the best of our knowledge.

- We propose three modules: disentangled causal subgraph identification, graph embedding intervention, and invariant architecture customization, offering a nuanced strategy for extracting and utilizing causal relationships between graph data and architecture, which is stable under distribution shifts, thereby enhancing the model's capability of out-of-distribution generalization.

- Extensive experiments on both synthetic and real-world datasets confirm that CARNAS significantly outperforms existing baselines, showcasing its efficacy in improving graph classification accuracy across diverse datasets, and validating the superior out-of-distribution generalization capabilities of our proposed CARNAS. [1]

---

[1]We provide the datasets and codes of our paper in the anonymous link.

## 2 Preliminary

### 2.1 Graph NAS under distribution shifts

Denote $\mathbb{G}$ and $\mathbb{Y}$ as the graph and label space. We consider a training graph dataset $\mathcal{G}_{tr} = \{(G_i, Y_i)\}_{i=1}^{N_{tr}}$ and a testing graph dataset $\mathcal{G}_{te} = \{(G_i, Y_i)\}_{i=1}^{N_{te}}$, where $G_i \in \mathbb{G}$, $Y_i \in \mathbb{Y}$, $N_{tr}$ and $N_{te}$ represent the number of graph instances in training set and testing set, respectively. The generalization of graph classification under distribution shifts can be formed as:

**Problem 1** *We aim to find the optimal prediction model $F^*(\cdot) : \mathbb{G} \to \mathbb{Y}$ that performs well on $\mathcal{G}_{te}$ when there is a distribution shift between training and testing data, i.e. $P(\mathcal{G}_{tr}) \neq P(\mathcal{G}_{te})$:*

$$F^*(\cdot) = \arg\min_F \mathbb{E}_{(G,Y)\sim P(\mathcal{G}_{te})} \left[ \ell(F(G), Y) \mid \mathcal{G}_{tr} \right], \tag{1}$$

*where $\ell(\cdot, \cdot) : \mathbb{Y} \times \mathbb{Y} \to \mathbb{R}$ is a loss function.*

Graph NAS methods search the optimal GNN architecture $A^*$ from the search space $\mathcal{A}$, and form the complete model $F$ together with the learnable parameters $\omega$. Unlike most existing works using a fixed GNN architecture for all graphs, (41) is the first to customize a GNN architecture for each graph, supposing that the architecture only depends on the graph. We follow the idea and inspect deeper concerning the graph neural architecture search process.

### 2.2 Causal view of the Graph NAS process

Causal approaches are largely adopted when dealing with out-of-distribution (OOD) generalization by capturing the stable causal structures or patterns in input data that influence the results (27). While in normal graph neural network cases, previous work that studies the problem from a causal perspective mainly considers the causality between graph data and labels (28; 53).

**Causal analysis in Graph NAS.** Based on the known that different GNN architectures suit different graphs (7; 60) and inspired by (56), we analyze the potential relationships between graph instance $G$, causal subgraph $G_c$, non-causal subgraph $G_s$ and optimal architecture $A^*$ for $G$ in the graph neural architecture search process as below:

- $G_c \to G \leftarrow G_s$ indicates that two disjoint parts, causal subgraph $G_c$ and non-causal subgraph $G_s$, together form the input graph $G$.
- $G_c \to A^*$ represents our assumption that there exists the causal subgraph which solely determines the optimal architecture $A^*$ for input graph $G$. Taking the Spurious-Motif dataset (63) as an example, (41) discovers that different shapes of graph elements prefer different architectures.
- $G_c \dashleftarrow\dashrightarrow G_s$ means that there are potential probabilistic dependencies between $G_c$ and $G_s$ (39; 40), which can make up spurious correlations between non-causal subgraph $G_s$ and the optimal architecture $A^*$.

**Intervention.** Inspired by the ideology of invariant learning (2; 24; 5), that forms different environments to abstract the invariant features, we do interventions on causal subgraph $G_c$ by adding different spurious (non-causal) subgraphs to it, and therefore simulate different environments for a graph instance $G$.

### 2.3 Problem formalization

Based on the above analysis, we propose to search for a causal-aware GNN architecture for each input graph. To be specific, we target to guide the search for the optimal architecture $A^*$ by identifying the causal subgraph $G_c$ in the Graph NAS process. Therefore, Problem 1 is transformed into the following concrete task as in Problem 2.

**Problem 2** *We systematize model $F : \mathbb{G} \to \mathbb{Y}$ into three modules, i.e. $F = f_C \circ f_A \circ f_Y$, in which $f_C(G) = G_c : \mathbb{G} \to \mathbb{G}_c$ abstracts the causal subgraph $G_c$ from input graph $G$, where causal subgraph space $\mathbb{G}_c$ is a subset of $\mathbb{G}$, $f_A(G_c) = A : \mathbb{G}_c \to \mathcal{A}$ customizes the GNN architecture $A$ for*

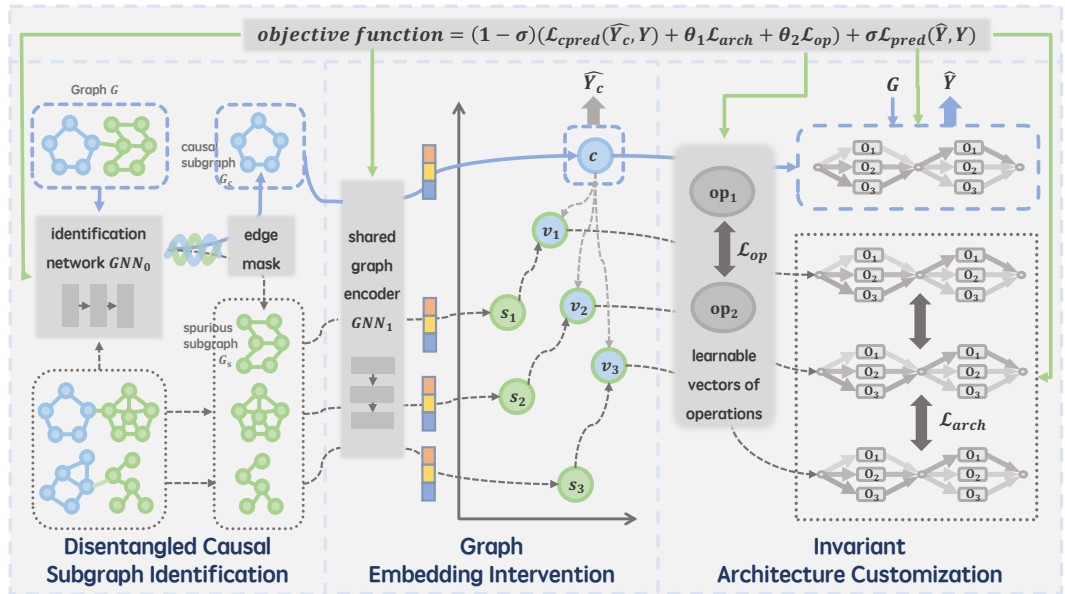

Figure 1: The framework of our proposed method **CARNAS**. As for an input graph $G$, the disentangled causal subgraph identification module abstracts its causal subgraph $G_c$ with disentangled GNN layers. Then, in the graph embedding intervention module, we conduct several interventions on $G_c$ with non-causal subgraphs in latent space and obtain $\mathcal{L}_{cpred}$ from the embedding of $G_c$ in the meanwhile. After that, the invariant architecture customization module aims to deal with distribution shift by customizing architecture from $G_c$ to attain $\hat{Y}$, $\mathcal{L}_{pred}$, and form $\mathcal{L}_{arch}$, $\mathcal{L}_{op}$ to further constrain the causal invariant property of $G_c$. Blue lines present the prediction approach and grey lines show other processes in the training stage. Additionally, green lines denote the updating process.

*causal subgraph $G_c$, and $f_Y(G, A) = \hat{Y} : \mathbb{G} \times \mathcal{A} \to \mathbb{Y}$ outputs the prediction $\hat{Y}$. Further, we derive the following objective function:*

$$\min_{f_C, f_A, f_Y} \sigma \mathcal{L}_{pred} + (1 - \sigma)\mathcal{L}_{causal}, \tag{2}$$

$$\mathcal{L}_{pred} = \sum_{i=1}^{N_{tr}} \ell \left( F_{f_C(G_i), f_A(G_{ci}), f_Y(G_i, A_i)}(G_i), Y_i \right), \tag{3}$$

*where $\mathcal{L}_{pred}$ guarantees the final prediction performance of the whole model, $\mathcal{L}_{causal}$ is a regularizer for causal constraints and $\sigma$ is the hyper-parameter to adjust the optimization of those two parts.*

## 3 METHOD

We present our proposed method in this section based on the above causal view. Firstly, we present the disentangled causal subgraph identification module to obtain the causal subgraph for searching optimal architecture in Section 3.1. Then, we propose the intervention module in Section 3.2, to help with finding the invariant subgraph that is causally correlated with the optimal architectures, making the NAS model intrinsically interpretable to some degree. In Section 3.3, we introduce the simulated customization module which aims to deal with distribution shift by customizing for each graph and simulating the situation when the causal subgraph is affected by different spurious parts. Finally, we show the total invariant learning and optimization procedure in Section 3.4.

### 3.1 DISENTANGLED CAUSAL SUBGRAPH IDENTIFICATION

This module utilizes disentangled GNN layers to capture different latent factors of the graph structure and further split the input graph instance $G$ into two subgraphs: causal subgraph $G_c$ and non-causal subgraph $G_s$. Specifically, considering an input graph $G = (\mathcal{V}, \mathcal{E})$, its adjacency matrix is

$\mathbf{D} \in \{0,1\}^{|\mathcal{V}| \times |\mathcal{V}|}$, where $\mathbf{D}_{i,j} = 1$ denotes that there exists an edge between node $V_i$ and node $V_j$, while $\mathbf{D}_{i,j} = 0$ otherwise. Since optimizing a discrete binary matrix $\mathbf{M} \in \{0,1\}^{|\mathcal{V}| \times |\mathcal{V}|}$ is unpractical due to the enormous number of subgraph candidates (63), and learning $\mathbf{M}$ separately for each input graph fails in generalizing to unseen test graphs (33), we adopt shared learnable disentangled GNN layers to comprehensively unveil the latent graph structural features and better abstract causal subgraphs. Firstly, we denote $Q$ as the number of latent features taken into account, and learn $Q$-chunk node representations by $Q$ GNNs:

$$\mathbf{Z}^{(l)} = \|_{q=1}^{Q} \mathrm{GNN}_0 \left( \mathbf{Z}_q^{(l-1)}, \mathbf{D} \right), \tag{4}$$

where $\mathbf{Z}_q^l$ is the $q$-th chunk of the node representation at $l$-th layer, $\mathbf{D}$ is the adjacency matrix, and $\|$ denotes concatenation. Then, we generate the edge importance scores $\mathcal{S}_\mathcal{E} \in \mathbb{R}^{|\mathcal{E}| \times 1}$ with an MLP:

$$\mathcal{S}_\mathcal{E} = \mathrm{MLP} \left( \mathbf{Z}_{row}^{(L)}, \mathbf{Z}_{col}^{(L)} \right), \tag{5}$$

where $\mathbf{Z}^{(L)} \in \mathbb{R}^{|\mathcal{V}| \times d}$ is the node representations after $L$ layers of disentangled GNN, and $\mathbf{Z}_{row}^{(L)}, \mathbf{Z}_{col}^{(L)}$ are the subsets of $\mathbf{Z}^{(L)}$ containing the representations of row nodes and column nodes of edges $\mathcal{E}$ respectively. After that, we attain the causal and non-causal subgraphs by picking out the important edges through $\mathcal{S}_\mathcal{E}$:

$$\mathcal{E}_c = \mathrm{Top}_t(\mathcal{S}_\mathcal{E}), \ \mathcal{E}_s = \mathcal{E} - \mathcal{E}_c, \tag{6}$$

where $\mathcal{E}_c$ and $\mathcal{E}_s$ denotes the edge sets of $G_c$ and $G_s$, respectively, and $\mathrm{Top}_t(\cdot)$ selects the top $t$-percentage of edges with the largest edge score values.

## 3.2 GRAPH EMBEDDING INTERVENTION

After obtaining the causal subgraph $G_c$ and non-causal subgraph $G_s$ of an input graph $G$, we use another shared $\mathrm{GNN}_1$ to encode those subgraphs so as to do interventions in the same latent space:

$$\mathbf{Z_c} = \mathrm{GNN}_1 \left( G_c \right), \ \mathbf{Z_s} = \mathrm{GNN}_1 \left( G_s \right). \tag{7}$$

Moreover, a readout layer is placed to aggregate node-level representations into graph-level representations:

$$\mathbf{H_c} = \mathrm{READOUT} \left( \mathbf{Z_c} \right), \ \mathbf{H_s} = \mathrm{READOUT} \left( \mathbf{Z_s} \right). \tag{8}$$

**Supervised classification for causal subgraphs.** We claim that the causal subgraph $G_c$ inferred in Section 3.1 for finding the optimal GNN architecture is supposed to contain the main characteristic of graph $G$'s structure as well as capture the essential part for the final graph classification predicting task. Hence, we employ a classifier on $\mathbf{H_c}$ to construct a supervised classification loss:

$$\mathcal{L}_{cpred} = \sum_{i=1}^{N_{tr}} \ell \left( \hat{Y}_{c_i}, Y_i \right), \ \hat{Y}_{c_i} = \Phi \left( \mathbf{H}_{\mathbf{c}i} \right), \tag{9}$$

where $\Phi$ is a classifier, $\hat{Y}_{c_i}$ is the prediction of graph $G_i$'s causal subgraph $G_{c_i}$ and $Y_i$ is the ground truth label of $G_i$.

**Interventions by non-causal subgraphs.** Based on subgraphs' embedding $\mathbf{H_c}$ and $\mathbf{H_s}$, we formulate the intervened embedding $\mathbf{H_v}$ in the latent space. Specifically, we collect all the representations of non-causal subgraphs $\{\mathbf{H}_{\mathbf{s}i}\}, i \in [1, N_{tr}]$, corresponding to each input graph $\{G_i\}, i \in [1, N_{tr}]$, in the current batch, and randomly sample $N_s$ of them as the candidates $\{\mathbf{H}_{\mathbf{s}j}\}, j \in [1, N_s]$ to do intervention with. As for a causal subgraph $G_c$ with representation $\mathbf{H_c}$, we define the representation under an intervention as:

$$do(S = G_{sj}): \ \mathbf{H}_{\mathbf{v}j} = (1 - \mu) \cdot \mathbf{H_c} + \mu \cdot \mathbf{H}_{\mathbf{s}j}, \ j \in [1, N_s], \tag{10}$$

in which $\mu \in (0, 1)$ is the hyper-parameter to control the intensity of an intervention.

## 3.3 INVARIANT ARCHITECTURE CUSTOMIZATION

After obtaining graph representations $\mathbf{H_c}$ and $\mathbf{H}_{\mathbf{v}j}, j \in [1, N_s]$, we introduce the method to construct a specific GNN architecture from a graph representation on the basis of differentiable NAS (31).

**Architecture customization.** To begin with, we denote the space of operator candidates as $\mathcal{O}$ and the number of architecture layers as $K$. Then, the ultimate architecture $A$ can be represented as a super-network:

$$g^k(\mathbf{x}) = \sum_{u=1}^{|\mathcal{O}|} \alpha_u^k o_u(\mathbf{x}),\ k \in [1, K],\tag{11}$$

where $\mathbf{x}$ is the input to layer $k$, $o_u(\cdot)$ is the operator from $\mathcal{O}$, $\alpha_u^k$ is the mixture coefficient of operator $o_u(\cdot)$ in layer $k$, and $g^k(x)$ is the output of layer $k$. Thereat, an architecture $A$ can be represented as a matrix $\mathbf{A} \in \mathbb{R}^{K \times |\mathcal{O}|}$, in which $\mathbf{A}_{k,u} = \alpha_u^k$. We learn these coefficients from graph representation $\mathbf{H}$ via trainable prototype vectors $\mathbf{op}_u^k$ ($u \in [1, |\mathcal{O}|], k \in [1, K]$), of operators:

$$\alpha_u^k = \frac{\exp\left(\mathbf{op}_u^{k\,T}\mathbf{H}\right)}{\sum_{u'=1}^{|\mathcal{O}|} \exp\left(\mathbf{op}_{u'}^{k\,T}\mathbf{H}\right)}.\tag{12}$$

In addition, the regularizer for operator prototype vectors:

$$\mathcal{L}_{op} = \sum_k \sum_{u,u' \in [1,|\mathcal{O}|], u \neq u'} \cos(\mathbf{op}_u^k, \mathbf{op}_{u'}^k),\tag{13}$$

where $\cos(\cdot, \cdot)$ is the cosine distance between two vectors, is engaged to avoid the mode collapse, following the exploration in (41).

**Architectures from causal subgraph and intervention graphs.** So far we form the mapping of $f_A : \mathbb{G} \to \mathcal{A}$ in Problem 2. As for an input graph $G$, we get its optimal architecture $A_c$ with the matrix $\mathbf{A_c}$ based on its causal subgraph's representation $\mathbf{H_c}$ through equation (12), while for each intervention graph we have $\mathbf{A}_{\mathbf{v}j}$ based on $\mathbf{H}_{\mathbf{v}j}$, $j \in [1, N_s]$ similarly.

The customized architecture $A_c$ is used to produce the ultimate prediction of input graph $G$ by $f_Y : \mathbb{G} \times \mathcal{A} \to \mathbb{Y}$ in Problem 2, and we formulate the main classification loss as:

$$\mathcal{L}_{pred} = \sum_{i=1}^{N_{tr}} \ell(\hat{Y}_i, Y_i),\ \hat{Y}_i = f_Y(G_i, A_{ci}).\tag{14}$$

Furthermore, we regard each $\mathbf{A}_{\mathbf{v}j}$, $j \in [1, N_s]$ as an outcome when causal subgraph $G_c$ is in a specific environment (treating the intervened part, i.e. non-causal subgraphs, as different environments). Therefore, the following variance regularizer is proposed as a causal constraint to compel the inferred causal subgraph $G_c$ to have the steady ability to solely determine the optimal architecture for input graph instance $G$:

$$\mathcal{L}_{arch} = \frac{1}{N_{tr}} \sum_{i=1}^{N_{tr}} \mathbf{1}^T \cdot \mathbf{Var}_i \cdot \mathbf{1},\ \mathbf{Var}_i = \mathrm{var}\left(\{\mathbf{A}_{\mathbf{v}ij}\}\right),\ j \in [1, N_s],\tag{15}$$

where $\mathrm{var}(\cdot)$ calculates the variance of a set of matrix, $\mathbf{1}^T \cdot \mathbf{Var}_i \cdot \mathbf{1}$ represents the summation of elements in matrix $\mathbf{Var}_i$.

### 3.4 OPTIMIZATION FRAMEWORK

Up to now, we have introduced $f_C : \mathbb{G} \to \mathbb{G}_c$ in section 3.1, $f_A : \mathbb{G}_c \to \mathcal{A}$ in section 3.2 and 3.3, $f_Y : \mathbb{G} \times \mathcal{A} \to \mathbb{Y}$ in section 3.3, and whereby deal with Problem 2. To be specific, the overall objective function in equation (2) is as below:

$$\mathcal{L}_{all} = \sigma\mathcal{L}_{pred} + (1 - \sigma)\mathcal{L}_{causal},\ \mathcal{L}_{causal} = \mathcal{L}_{cpred} + \theta_1\mathcal{L}_{arch} + \theta_2\mathcal{L}_{op},\tag{16}$$

where $\theta_1$, $\theta_2$ and $\sigma$ are hyper-parameters. Additionally, we adopt a linearly growing $\sigma_p$ corresponding to the epoch number $p$ as:

$$\sigma_p = \sigma_{min} + (p - 1)\frac{\sigma_{max} - \sigma_{min}}{P},\ p \in [1, P],\tag{17}$$

where $P$ is the maximum number of epochs. In this way, we can dynamically adjust the training key point in each epoch by focusing more on the causal-aware part (i.e. identifying suitable causal subgraph and learning vectors of operators) in the early stages and focusing more on the performance of the customized super-network in the later stages. We show the dynamic training process and how $\sigma_p$ improve the training and convergence efficiency in Appendix E.3. The overall framework and optimization procedure of the proposed CARNAS are summarized in Figure 1 and Algorithm 1.

## 4 EXPERIMENTS

In this section, we present the comprehensive results of our experiments on both synthetic and real-world datasets to validate the effectiveness of our approach. We also conduct a series of ablation studies to thoroughly examine the contribution of the components within our framework. More analysis on experimental results, training efficiency, complexity and sensitivity are in Appendix E.

### 4.1 EXPERIMENT SETTING

**Setting.** To ensure reliability and reproducibility, we execute each experiment ten times using distinct random seeds and present the average results along with their standard deviations. We do not employ validation dataset for conducting architecture search. The configuration and use of datasets in our method align with those in other GNN methods, ensuring fairness across all approaches.

**Baselines.** We compare our model with 12 baselines from the following two different categories:

- **Manually design GNNs.** We incorporate widely recognized architectures GCN (22), GAT (49), GIN (59), SAGE (15), and GraphConv (37), and MLP into our search space as **candidate operators** as well as baseline methods in our experiments. Apart from that, we include two recent advancements: ASAP (42) and DIR (56), which is specifically proposed for out-of-distribution generalization.

- **Graph Neural Architecture Search.** For **classic NAS**, we compare with DARTS (31), a differentiable architecture search method, and random search. For **graph NAS**, we explore a reinforcement learning-based

Table 1: The test accuracy of all methods on synthetic dataset Spurious-Motif. Values after $\pm$ denote the standard deviations. The best results overall are in bold and the best results of baselines in each category are underlined separately.

| Method | $b = 0.7$ | $b = 0.8$ | $b = 0.9$ |
|--------|-----------|-----------|-----------|
| GCN | $48.39_{\pm 1.69}$ | $41.55_{\pm 3.88}$ | $39.13_{\pm 1.76}$ |
| GAT | $50.75_{\pm 4.89}$ | $42.48_{\pm 2.46}$ | $40.10_{\pm 5.19}$ |
| GIN | $36.83_{\pm 5.49}$ | $34.83_{\pm 3.10}$ | $37.45_{\pm 3.59}$ |
| SAGE | $46.66_{\pm 2.51}$ | $44.50_{\pm 5.79}$ | $44.79_{\pm 4.83}$ |
| GraphConv | $47.29_{\pm 1.95}$ | $44.67_{\pm 5.88}$ | $44.82_{\pm 4.84}$ |
| MLP | $48.27_{\pm 1.27}$ | $46.73_{\pm 3.48}$ | $\underline{46.41_{\pm 2.34}}$ |
| ASAP | $\underline{54.07_{\pm 13.85}}$ | $\underline{48.32_{\pm 12.72}}$ | $43.52_{\pm 8.41}$ |
| DIR | $50.08_{\pm 3.46}$ | $48.22_{\pm 6.27}$ | $43.11_{\pm 5.43}$ |
| Random | $45.92_{\pm 4.29}$ | $51.72_{\pm 5.38}$ | $45.89_{\pm 5.09}$ |
| DARTS | $50.63_{\pm 8.90}$ | $45.41_{\pm 7.71}$ | $44.44_{\pm 4.42}$ |
| GNAS | $55.18_{\pm 18.62}$ | $51.64_{\pm 19.22}$ | $37.56_{\pm 5.43}$ |
| PAS | $52.15_{\pm 4.35}$ | $43.12_{\pm 5.95}$ | $39.84_{\pm 1.67}$ |
| GRACES | $65.72_{\pm 17.47}$ | $59.57_{\pm 17.37}$ | $50.94_{\pm 8.14}$ |
| DCGAS | $\underline{87.68_{\pm 6.12}}$ | $\underline{75.45_{\pm 17.40}}$ | $\underline{61.42_{\pm 16.26}}$ |
| **CARNAS** | $\mathbf{94.41_{\pm 4.58}}$ | $\mathbf{88.04_{\pm 13.77}}$ | $\mathbf{87.15_{\pm 11.85}}$ |

method GNAS (11), and PAS (51) that is specially designed for graph classification tasks. Additionally, we compare two state-of-the-art graph NAS methods that are specially designed for non-i.i.d. graph datasets, including GRACES (41) and DCGAS (61).

### 4.2 ON SYNTHETIC DATASETS

**Datasets.** The synthetic dataset, **Spurious-Motif** (41; 56; 63), encompasses 18,000 graphs, each uniquely formed by combining a base shape (denoted as *Tree*, *Ladder*, or *Wheel* with $S = 0, 1, 2$) with a motif shape (represented as *Cycle*, *House*, or *Crane* with $C = 0, 1, 2$). Notably, the classification of each graph relies solely on its motif shape, despite the base shape typically being larger. This dataset is particularly designed to study the effect of distribution shifts, with a distinct bias introduced solely on the training set through the probability distribution $P(S) = b \times \mathbb{I}(S = C) + \frac{1-b}{2} \times \mathbb{I}(S \neq C)$, where $b$ modulates the correlation between base and motif shapes, thereby inducing a deliberate shift between the training set and testing set, where all base and motif shapes are independent with equal

probabilities. We choose $b = 0.7/0.8/0.9$, enabling us to explore our model's performance under various significant distributional variations. The effectiveness of our approach is measured using **accuracy** as the evaluation metric on this dataset.

**Results.** Table 1 presents the experimental results on three synthetic datasets, revealing that our model significantly outperforms all baseline models across different scenarios.

Specifically, we observe that the performance of all GNN models is particularly poor, suggesting their sensitivity to spurious correlations and their inability to adapt to distribution shifts. However, DIR (56), designed specifically for non-I.I.D. datasets and focusing on discovering invariant rationale to enhance generalizability, shows pretty well performance compared to most of the other GNN models. This reflects the feasibility of employing causal learning to tackle generalization issues.

Moreover, NAS methods generally yield slightly better outcomes than manually designed GNNs in most scenarios, emphasizing the significance of automating architecture by learning the correlations between input graph data and architecture to search for the optimal GNN architecture. Notably, methods specifically designed for non-I.I.D. datasets, such as GRACES (41), DCGAS (61), and our CARNAS, exhibit significantly less susceptibility to distribution shifts compared to NAS methods intended for I.I.D. data.

Among these, our approach consistently achieves the best performance across datasets with various degrees of shifts, demonstrating the effectiveness of our method in enhancing Graph NAS performance, especially in terms of out-of-distribution generalization, which is attained by effectively capturing causal invariant subgraphs to guide the architecture search process, and filtering out spurious correlations meanwhile.

### 4.3 ON REAL-WORLD DATASETS

**Datasets.** The real-world datasets **OGBG-Mol\***, including Ogbg-molhiv, Ogbg-molbace, and Ogbg-molsider (16; 58), feature 41127, 1513, and 1427 molecule graphs, respectively, aimed at molecular property prediction. The division of the datasets is based on scaffold values, designed to segregate molecules according to their structural frameworks, thus introducing a significant challenge to the prediction of graph properties. The predictive performance of our approach across these diverse molecular structures and properties is measured using **ROC-AUC** as the evaluation metric.

Table 2: The test ROC-AUC of all methods on real-world datasets OGBG-Mol\*. Values after $\pm$ denote the standard deviations. The best results overall are in bold and the best results of baselines in each category are underlined separately.

| Method | HIV | SIDER | BACE |
|---|---|---|---|
| GCN | $75.99_{\pm 1.19}$ | $\underline{59.84}_{\pm 1.54}$ | $68.93_{\pm 6.95}$ |
| GAT | $76.80_{\pm 0.58}$ | $57.40_{\pm 2.01}$ | $75.34_{\pm 2.36}$ |
| GIN | $\underline{77.07}_{\pm 1.49}$ | $57.57_{\pm 1.56}$ | $73.46_{\pm 5.24}$ |
| SAGE | $75.58_{\pm 1.40}$ | $56.36_{\pm 1.32}$ | $74.85_{\pm 2.74}$ |
| GraphConv | $74.46_{\pm 0.86}$ | $56.09_{\pm 1.06}$ | $\underline{78.87}_{\pm 1.74}$ |
| MLP | $70.88_{\pm 0.83}$ | $58.16_{\pm 1.41}$ | $71.60_{\pm 2.30}$ |
| ASAP | $73.81_{\pm 1.17}$ | $55.77_{\pm 1.18}$ | $71.55_{\pm 2.74}$ |
| DIR | $77.05_{\pm 0.57}$ | $57.34_{\pm 0.36}$ | $76.03_{\pm 2.20}$ |
| DARTS | $74.04_{\pm 1.75}$ | $60.64_{\pm 1.37}$ | $76.71_{\pm 1.83}$ |
| PAS | $71.19_{\pm 2.28}$ | $59.31_{\pm 1.48}$ | $76.59_{\pm 1.87}$ |
| GRACES | $77.31_{\pm 1.00}$ | $61.85_{\pm 2.58}$ | $79.46_{\pm 3.04}$ |
| DCGAS | $\underline{78.04}_{\pm 0.71}$ | $\underline{63.46}_{\pm 1.42}$ | $\underline{81.31}_{\pm 1.94}$ |
| **CARNAS** | $\mathbf{78.33}_{\pm 0.64}$ | $\mathbf{83.36}_{\pm 0.62}$ | $\mathbf{81.73}_{\pm 2.92}$ |

**Results.** Results from real-world datasets are detailed in Table 2, where our CARNAS model once again surpasses all baselines across three distinct datasets, showcasing its ability to handle complex distribution shifts under various conditions.

For manually designed GNNs, the optimal model varies across different datasets: GIN achieves the best performance on Ogbg-molhiv, GCN excels on Ogbg-molsider, and GraphConv leads on Ogbg-molbace. This diversity in performance confirms a crucial hypothesis in our work, that different GNN models are predisposed to perform well on graphs featuring distinct characteristics.

In the realm of NAS models, we observe that DARTS and PAS, proposed for I.I.D. datasets, perform comparably to manually crafted GNNs, whereas GRACES, DCGAS and our CARNAS, specifically designed for non-I.I.D. datasets outshine other baselines. Our approach reaches the top performance

across all datasets, with a particularly remarkable breakthrough on Ogbg-molsider, highlighting our method's superior capability in adapting to and excelling within diverse data environments.

## 4.4 ABLATION STUDY

In this section, we conduct ablation studies to examine the effectiveness of each vital component in our framework. Detailed settings are in Appendix D.3.1.

**Results.** From Figure 2, we have the following observations. First of all, our proposed CARNAS outperforms all the variants as well as the best-performed baseline on all datasets, demonstrating the effectiveness of each component of our proposed method. Secondly, the performance of 'CARNAS w/o $\mathcal{L}_{arch}$', 'CARNAS w/o $\mathcal{L}_{cpred}$' and 'CARNAS w/o $\mathcal{L}_{arch}$ & $\mathcal{L}_{cpred}$' dropped obviously on all datasets comparing with the full CARNAS , which validates that our proposed modules help the model to identify stable causal components from comprehensive graph feature and further guide the Graph NAS process to enhance its performance significantly especially under distribution shifts. What's more, though 'CARNAS w/o $\mathcal{L}_{arch}$' decreases, its performance still surpasses the best results in baselines across all datasets, indicating that even if the invariance of the influence of the causal subgraph on the architecture is

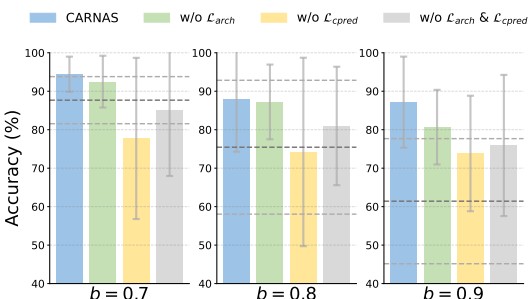

Figure 2: Results of ablation studies on synthetic datasets, where 'w/o $\mathcal{L}_{arch}$' removes $\mathcal{L}_{arch}$ from the overall loss in Eq. (16), 'w/o $\mathcal{L}_{cpred}$' removes $\mathcal{L}_{cpred}$, and 'w/o $\mathcal{L}_{arch}$ & $\mathcal{L}_{cpred}$' removes both of them. The error bars report the standard deviations. Besides, the average and standard deviations of the best-performed baseline on each dataset are denoted as the dark and light thick dash lines respectively.

not strictly restricted by $\mathcal{L}_{arch}$, it is effective to use merely the causal subgraph guaranteed by $\mathcal{L}_{cpred}$ to contain the important information of the input graph and use it to guide the architecture search.

## 5 RELATED WORK

Neural Architecture Search (NAS) automates creating optimal neural networks using RL-based (76), evolutionary (43), and gradient-based methods (31). GraphNAS (11) inspired studies on GNN architectures for graph classification in various datasets (41; 61). Real-world data differences between training and testing sets impact GNN performance (46; 27; 70; 71). Studies (28; 9) identify invariant subgraphs to mitigate this, usually with fixed GNN encoders. Our method automates designing generalized graph architectures by discovering causal relationships. Causal learning explores variable interconnections (40), enhancing deep learning (65). In graphs, it includes interventions on non-causal components (57), causal and bias subgraph decomposition (9), and ensuring out-of-distribution generalization (6; 72; 73). These methods use fixed GNN architectures, while we address distribution shifts by discovering causal relationships between graphs and architectures.

## 6 CONCLUSION

In this paper, we address distribution shifts in graph neural architecture search (Graph NAS) from a causal perspective. Existing methods struggle with distribution shifts between training and testing sets due to spurious correlations. To mitigate this, we introduce Causal-aware Graph Neural Architecture Search (**CARNAS**). Our approach identifies stable causal structures and their relationships with architectures. We propose three key modules: Disentangled Causal Subgraph Identification, Graph Embedding Intervention, and Invariant Architecture Customization. These modules leverage causal relationships to search for generalized graph neural architectures. Experiments on synthetic and real-world datasets show that CARNAS achieves superior out-of-distribution generalization, highlighting the importance of causal awareness in Graph NAS.

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

# A  NOTATION

Table 3: Important Notation

| Notation | Meaning |
|---|---|
| $\mathbb{G}, \mathbb{Y}$ | Graph space and Label space |
| $\mathcal{G}_{tr}, \mathcal{G}_{te}$ | Training graph dataset, Testing graph dataset |
| $G_i, Y_i$ | Graph instance $i$, Label of graph instance $i$ |
| $G_c, G_s$ | Causal subgraph and Non-causal subgraph |
| $\mathbf{D}$ | Adjacency matrix of graph $G$ |
| $\mathbf{Z}$ | Node representations |
| $\mathcal{S}_{\mathcal{E}}$ | Edge importance scores |
| $\mathcal{E}_c, \mathcal{E}_s$ | Edge set of the causal subgraph and the non-causal subgraph |
| $\mathbf{H_c}, \mathbf{H_s}$ | Graph-level representation of causal subgraph and non-causal subgraph |
| $\hat{Y}_{c_i}$ | Prediction of graph $G_i$'s causal subgraph $G_{c_i}$ |
| $\mathbf{H_v}$ | Intervened graph representation |
| $\mathcal{O}, o_u(\cdot)$ | Space of operator candidates, operator from $\mathcal{O}$ |
| $K$ | Number of architecture layers |
| $\mathcal{A}$ | Search space of GNN architectures |
| $A, \mathbf{A}$ | An architecture represented as a super-network; matrix of architecture A |
| $g^k(\mathbf{x})$ | Output of layer $k$ in the architecture. |
| $\alpha_u^k$ | Mixture coefficient of operator $o_u(\cdot)$ in layer $k$ |
| $\mathbf{op}_u^k$ | Trainable prototype vectors of operators |

# B  ALGORITHM

The overall framework and optimization procedure of the proposed CARNAS are summarized in Figure 1 and Algorithm 1, respectively.

---
**Algorithm 1** The overall algorithm of **CARNAS**
---

**Require:** Training Dataset $\mathcal{G}_{tr}$,
    Hyper-parameters $t$ in Eq. (6), $\mu$ in Eq. (10), $\theta_1, \theta_2$ in Eq. (16)
  1: Initialize all trainable parameters
  2: **for** $p = 1, \ldots, P$ **do**
  3:     Set $\sigma_p$ as Eq. (17)
  4:     Derive causal and non-causal subgraphs as Eq. (4) (5) (6)
  5:     Calculate graph representations of causal and non-causal subgraphs as Eq. (7) (8)
  6:     Calculate $\mathcal{L}_{cpred}$ using Eq. (9)
  7:     Sample $N_s$ non-causal subgraphs as candidates
  8:     **for** causal subgraph $G_c$ of graph $G$ in $\mathcal{G}_{tr}$ **do**
  9:         Do interventions on $G_c$ in latent space as Eq. (10)
10:         Calculate architecture matrix $\mathbf{A_c}$ and $\{\mathbf{A_v}_j\}$ from causal subgraph and their intervention graphs as Eq. (12)
11:     **end for**
12:     Calculate $\mathcal{L}_{op}$ using Eq. (13)
13:     Calculate $\mathcal{L}_{pred}$ using Eq. (11) (14)
14:     Calculate $\mathcal{L}_{arch}$ using Eq. (15))
15:     Calculate the overall loss $\mathcal{L}_{all}$ using Eq. (16)
16:     Update parameters using gradient descends
17: **end for**

---

# C  THEORETICAL ANALYSIS

In this section, in order to more rigorously establish our method, we provide a theoretical analysis about the problem of identifying and leveraging causal graph-architecture relationship to find the optimal architecture.

To begin with, since causal relationships are, by definition, invariant across environments, we make the below assumption on our causal invariant subgraph generator $f_C(G) = G_c : \mathbb{G} \to \mathbb{G}_c$, following previous literature on invariant learning (44; 28).

**Assumption 1** *There exists an optimal causal invariant subgraph generator $f_C(G)$ satisfying:*

    *a. **Invariance property:** For all $e, e' \in supp(\mathcal{E})$, $P^e(A^*|f_C(G)) = P^{e'}(A^*|f_C(G))$.*

    *b. **Sufficiency property:** $A^* = f_A(f_C(G)) + \epsilon$, where $f_A(\cdot)$ customizes the GNN architecture from a graph, $\epsilon \perp G$ (indicating statistical independence), and $\epsilon$ is random noise.*

Invariance assumption indicating that the subgraph generator $f_C(G)$ is capable of generating invariant subgraphs across different environments $e, e' \in supp(\mathcal{E})$, where $\mathcal{E}$ is a random variable of all environments. This ensures that the conditional distribution $P(A^*|f_C(G))$ remains consistent and unaffected by the environment. Sufficiency assumption demonstrates that the subgraph generated by $f_C(G)$ has sufficient expressive power to enable prediction of the optimal architecture $A^*$. This is achieved through $f_A(\cdot)$, customizing the GNN architecture from a graph, while the added random noise $\epsilon$ is independent of the graph $G$.

Then, how can we get the optimal causal invariant subgraph generator? Following previous work(28), we can prove that it can be obtain through maximizing $I(A^*; f_C(G))$, i.e. the mutual information between optimal architecture and the generated subgraph.

**Theorem 1 (Optimal Generator of Causal Subgraphs)** *A generator $f_C(G)$ is the optimal generator that satisfies Assumption 1 if and only if it is the maximal causal subgraph generator, i.e.,*

$$f_C^* = \arg \max_{f_C \in \mathcal{F}_\mathcal{E}} I(A^*; f_C(G)), \tag{18}$$

*where $\mathcal{F}_\mathcal{E}$ is the subgraph generator set with related to the random vector of all environments, and $I(\cdot; \cdot)$ is the mutual information between the optimal architecture $A^*$ and the generated causal subgraph.*

**Proof.** Let $\hat{f}_C = \arg \max_{f_C \in \mathcal{F}_\mathcal{E}} I(A^*; f_C(G))$. From the invariance property in Assumption 1, it follows that $f_C^* \in \mathcal{F}_\mathcal{E}$. To prove the theorem, we show that: $I(A^*; \hat{f}_C(G)) \leq I(A^*; f_C^*(G))$, which implies $\hat{f}_C = f_C^*$. Using the functional representation lemma (8), any random variable $X_2$ can be expressed as a function of another random variable $X_1$ and an independent random variable $X_3$. Applying this to $f_C^*(G)$ and $\hat{f}_C(G)$, there exists a $f_C'(G)$ such that $f_C'(G) \perp f_C^*(G)$ and $\hat{f}_C(G) = \gamma(f_C^*(G), f_C'(G))$, where $\gamma(\cdot)$ is a deterministic function. Then, the mutual information can be decomposed as follows:

$$\begin{aligned}
I(A^*; \hat{f}_C(G)) &= I(A^*; \gamma(f_C^*(G), f_C'(G))) \\
&\leq I(A^*; f_C^*(G), f_C'(G)) \\
&= I(f_A(f_C^*(G)); f_C^*(G), f_C'(G)) \\
&= I(f_A(f_C^*(G)); f_C^*(G)) \\
&= I(A^*; f_C^*(G)),
\end{aligned} \tag{19}$$

which completes the proof.

Since maximizing $I(A^*; f_C(G))$ is difficult, we transform it into anther way which will be introduced in later passage.

Next, we show the theorem that guide us to optimize the model, represented as $Q$, that can construct optimal architecture $A^*$ for graph instance $G$ under distribution shifts. The prediction is based on the causal subgraph $G_c^*$, which delineates the causal graph-architecture relationship. We denote the conditional distribution modeled by $Q$ as $q(A^*|G_c^*)$.

**Theorem 2** *Let $f_C^*$ be the optimal causal invariant subgraph generator from Assumption 1, and let $G_c^* = f_C^*(G)$ and $G_s^* = G \setminus G_c^*$. Then, we can get the optimal model $Q$ under distribution shifts by minimizing the following objective:*

$$\min \mathbb{E} \left[ \log \frac{p(A^*|G_c^*)}{q(A^*|G_c^*)} \right] + I(G_s^*; A^*|G_c^*). \tag{20}$$

*Here, $I(G_s^*; A^*|G_c^*)$ quantifies the spurious correlation between $G_s^*$ and $A^*$, which the model need to ignore, and the first term ensures that $q(A^*|G_c^*)$ closely matches $p(A^*|G_c^*)$.*

**Proof.**

From the sufficiency assumption of $f_C^*$ in Assumption 1, we know that: $A^* = f_A(f_C(G)) + \epsilon$, where $\epsilon \perp G$. This implies that $A^*$ is conditionally independent of $G_s^*$ (i.e., the non-causal subgraph) given $G_c^*$. Therefore, the full graph $G = (G_c^*, G_s^*)$ satisfies:

$$P(A^*|G) = P(A^*|G_c^*). \tag{21}$$

Additionally, by the invariance property, for any $e, e' \in \text{supp}(\mathcal{E})$, the conditional distribution of $A^*$ given $G_c^*$ remains invariant across environments: $P^e(A^*|G_c^*) = P^{e'}(A^*|G_c^*)$. This invariance guarantees that $Q$ will generalize well under distribution shifts caused by changes in the environment, when $q(A^*|G_c^*)$ approximates the stable $p(A^*|G_c^*)$.

To approximate $p(A^*|G_c^*)$ with $q(A^*|G_c^*)$, we minimize the negative conditional log-likelihood of the observed data:

$$-\ell = - \sum_{i=1}^{n} \log q(A_i^*|G_{c_i}^*). \tag{22}$$

Expanding this objective using $G = (G_c^*, G_s^*)$, we rewrite it as:

$$-\ell = \sum_{i=1}^{n} \log \frac{p(A_i^*|G_{c_i}^*)}{q(A_i^*|G_{c_i}^*)} + \sum_{i=1}^{n} \log \frac{p(A_i^*|G_i)}{p(A_i^*|G_{c_i}^*)} - \sum_{i=1}^{n} \log p(A_i^*|G_i) \tag{23}$$

$$= \mathbb{E} \left[ \log \frac{p(A^*|G_c^*)}{q(A^*|G_c^*)} \right] + \mathbb{E} \left[ \log \frac{p(A^*|G)}{p(A^*|G_c^*)} \right] - \mathbb{E} \left[ \log p(A^*|G) \right]. \tag{24}$$

The third term is irreducible constant inherent in the dataset, so we omit it when optimizing. Then, we decompose $G$ into $(G_c^*, G_s^*)$ and rewrite the second term as:

$$\mathbb{E} \left[ \log \frac{p(A^*|G)}{p(A^*|G_c^*)} \right] = \mathbb{E} \left[ \log \frac{p(A^*|G_c^*, G_s^*)}{p(A^*|G_c^*)} \right] \tag{25}$$

$$= \sum_{i=1}^{n} p(G_{c_i}^*, G_{s_i}^*, A_i^*) \log \frac{p(A_i^*|G_{c_i}^*, G_{s_i}^*)}{p(A_i^*|G_{c_i}^*)} \tag{26}$$

$$= \sum_{i=1}^{n} p(G_{c_i}^*, G_{s_i}^*, A_i^*) \log \frac{p(A_i^*|G_{c_i}^*, G_{s_i}^*) p(G_{s_i}^*|G_{c_i}^*)}{p(A_i^*|G_{c_i}^*) p(G_{s_i}^*|G_{c_i}^*)} \tag{27}$$

$$= \sum_{i=1}^{n} p(G_{c_i}^*, G_{s_i}^*, A_i^*) \log \frac{p(G_{s_i}^*, A_i^*|G_{c_i}^*)}{p(A_i^*|G_{c_i}^*) p(G_{s_i}^*|G_{c_i}^*)} \tag{28}$$

$$= I(G_s^*; A^*|G_c^*). \tag{29}$$

Thus, the final objective to optimize $q(A^*|G_c^*)$ is:

$$\min \mathbb{E} \left[ \log \frac{p(A^*|G_c^*)}{q(A^*|G_c^*)} \right] + I(G_s^*; A^*|G_c^*), \tag{30}$$

where the second term, $I(G_s^*; A^*|G_c^*)$, measures the residual spurious correlation between $G_s^*$ and $A^*$ given $G_c^*$.

This concludes the proof.

However, this objective is challenging to optimize directly in practice. To address this, we analyze each term intuitively and explain how our method is derived from the theorem.

The first term, $\mathbb{E}\left[\log \frac{p(A^*|G_c^*)}{q(A^*|G_c^*)}\right]$, ensures that the model accurately approximates the true conditional distribution $p(A^*|G_c^*)$ based on the causal subgraph $G_c^*$. Since the optimal architecture $A^*$ is defined as the one achieving the best predictive performance on label $Y$, we indirectly optimize the first term $\mathbb{E}\left[\log \frac{p(A^*|G_c^*)}{q(A^*|G_c^*)}\right]$ by focusing on label's prediction performance. Specifically, we minimize $\mathcal{L}_{pred}$ (Equation 14), which measures the loss between the ground-truth label and the prediction from the learned optimal architecture $A^*/A_c$. This surrogate loss guides $q(A^*|G_c^*)$ to approximate $p(A^*|G_c^*)$, as $A^*$ is inherently tied to the optimal predictive performance on final task.

The second term $I(G_s^*; A^*|G_c^*)$ represents the conditional mutual information between the optimal architecture $A^*$ and the spurious subgraph $G_s^*$, given the causal subgraph $G_c^*$. Minimizing this term encourages the model to reduce its reliance on the spurious subgraph $G_s^*$ when predicting the optimal architecture, given $G_c^*$. This motivates the use of $\mathcal{L}_{arch}$ in Equation 15, which measures the variance of simulated architectures corresponding to intervention graphs formed by combining the causal subgraph with different spurious subgraphs. By reducing this variance, the model is encouraged to rely solely on the causal subgraph $G_c^*$ for determining the optimal architecture, ensuring that the causal subgraph has a stable and consistent predictive capability across varying spurious components in input graph $G$. Then, we prove that $I(G_s^*; A^*|G_c^*) = I(G; A^*) - I(G_c^*; A^*)$:

**Proof.** By the chain rule of mutual information, we have

$$I(G; A^*) = I(G_c^*, G_s^*; A^*) = I(G_c^*; A^*) + I(G_s^*; A^*|G_c^*), \tag{31}$$

where $G = (G_c^*, G_s^*)$. Rearranging the equation, we obtain

$$I(G_s^*; A^*|G_c^*) = I(G; A^*) - I(G_c^*; A^*). \tag{32}$$

Thus, minimizing $I(G_s^*; A^*|G_c^*)$ in turn encourages maximizing $I(A^*; f_C(G))$, which proved to lead to optimizing the causal subgraph generator in Theorem 1.

Therefore, we propose to **jointly optimize causal graph-architecture relationship and architecture search** by offering an **end-to-end training strategy** for extracting and utilizing causal relationships between graph data and architecture, which is stable under distribution shifts, during the architecture search process, thereby enhancing the model's capability of OOD generalization.

# D    REPRODUCIBILITY DETAILS

## D.1    DEFINITION OF SEARCH SPACE

The number of layers in our model is predetermined before training, and the type of operator for each layer can be selected from our defined operator search space $\mathcal{O}$. We incorporate widely recognized architectures GCN, GAT, GIN, SAGE, GraphConv, and MLP into our search space as candidate operators in our experiments. This allows for the combination of various sub-architectures within a single model, such as using GCN in the first layer and GAT in the second layer. Furthermore, we consistently use standard global mean pooling at the end of the GNN architecture to generate a global embedding.

## D.2    DATASETS DETAILS

We utilize synthetic SPMotif datasets, which are characterized by three distinct degrees of distribution shifts, and three different real-world datasets, each with varied components, following previous works (41; 61; 56). Based on the statistics of each dataset as shown in Table 4, we conducted a comprehensive comparison across *various scales and graph sizes*. This approach has empirically validated the scalability of our model.

**Detailed description for real-world datasets**    The real-world datasets are 3 molecular property prediction datasets in OGB (16), and are adopted from the MoleculeNet (58). Each graph represents a molecule, where nodes are atoms, and edges are chemical bonds.

Table 4: Statistics for different datasets.

|  | Graphs | Avg. Nodes | Avg. Edges |
|---|---|---|---|
| ogbg-molhiv | 41127 | 25.5 | 27.5 |
| ogbg-molsider | 1427 | 33.6 | 35.4 |
| ogbg-molbace | 1513 | 34.1 | 36.9 |
| SPMotif-0.7/0.8/0.9 | 18000 | 26.1 | 36.3 |

- The HIV dataset was introduced by the Drug Therapeutics Program (DTP) AIDS Antiviral Screen, which tested the ability to inhibit HIV replication for over 40000 compounds. Screening results were evaluated and placed into 2 categories: inactive (confirmed inactive CI) and active (confirmed active CA and confirmed moderately active CM).

- The Side Effect Resource (SIDER) is a database of marketed drugs and adverse drug reactions (ADR). The version of the SIDER dataset in DeepChem has grouped drug side-effects into 27 system organ classes following MedDRA classifications measured for 1427 approved drugs (following previous usage).

- The BACE dataset provides quantitative ($IC_{50}$) and qualitative (binary label) binding results for a set of inhibitors of human $\beta$-secretase 1 (BACE-1). It merged a collection of 1522 compounds with their 2D structures and binary labels in MoleculeNet, built as a classification task.

The division of the datasets is based on scaffold values, designed to segregate molecules according to their structural frameworks, thus introducing a significant challenge to the prediction of graph properties.

### D.3 DETAILED HYPER-PARAMETER SETTINGS

We fix the number of latent features $Q = 4$ in Eq. (4), number of intervention candidates $N_s$ as batch size in Eq. (10), $\sigma_{min} = 0.1$, $\sigma_{max} = 0.7$, $P = 100$ in Eq. (17), and the tuned hyper-parameters for each dataset are as in Table 5.

Table 5: Hyper-parameter settings

| Dataset | $t$ in Eq. (6) | $\mu$ in Eq. (10) | $\theta_1$ in Eq. (16) | $\theta_2$ in Eq. (16) |
|---|---|---|---|---|
| SPMotif-0.7/0.8/0.9 | 0.85 | 0.26 | 0.36 | 0.010 |
| ogbg-molhiv | 0.46 | 0.68 | 0.94 | 0.007 |
| ogbg-molsider | 0.40 | 0.60 | 0.85 | 0.005 |
| ogbg-molbace | 0.49 | 0.54 | 0.80 | 0.003 |

#### D.3.1 DETAILED SETTINGS FOR ABLATION STUDY

We compare the following ablated variants of our model in Section 4.4:

- 'CARNAS w/o $\mathcal{L}_{arch}$' removes $\mathcal{L}_{arch}$ from the overall loss in Eq. (16). In this way, the contribution of the *graph embedding intervention module* together with the *invariant architecture customization module* to improve generalization performance by restricting the causally invariant nature for constructing architectures of the causal subgraph is removed.

- 'CARNAS w/o $\mathcal{L}_{cpred}$' removes $\mathcal{L}_{cpred}$, thereby relieving the supervised restriction on causal subgraphs for encapsulating sufficient graph features, which is contributed by *disentangled causal subgraph identification module* together with the *graph embedding intervention module* to enhance the learning of causal subgraphs.

- 'CARNAS w/o $\mathcal{L}_{arch}$ & $\mathcal{L}_{cpred}$' further removes both of them.

# E DEEPER ANALYSIS

## E.1 SUPPLEMENTARY ANALYSIS OF THE EXPERIMENTAL RESULTS

**Sythetic datasets.** We notice that the performance of CARNAS is way better than DIR (56), which also introduces causality in their method, on synthetic datasets. We provide an explanation as follows: Our approach differs from and enhances upon DIR in several key points. Firstly, unlike DIR, which uses normal GNN layers for embedding nodes and edges to derive a causal subgraph, we employ disentangled GNN. This allows for more effective capture of latent features when extracting causal subgraphs. Secondly, while DIR focuses on the causal relationship between a graph instance and its label, our study delves into the causal relationship between a graph instance and its optimal architecture, subsequently using this architecture to predict the label. Additionally, we incorporate NAS method, introducing an invariant architecture customization module, which considers the impact of architecture on performance. Based on these advancements, our method may outperform DIR.

**Real-world datasets.** We also notice that our methods improves a lot on the performance for the second real-world dataset SIDER. We further conduct an ablation study on SIDER to confirm that each proposed component contributes to its performance, as present in Figure 3. The model 'w/o $Larch$' shows a slight decrease in performance, while 'w/o $Lcpred$' exhibits a substantial decline. This indicates that both restricting the invariance of the influence of the causal subgraph on the architecture via $Larch$, and ensuring that the causal subgraph retains crucial information from the input graph via $Lcpred$, are vital for achieving high performance on SIDER, especially the latter which empirically proves to be exceptionally effective.

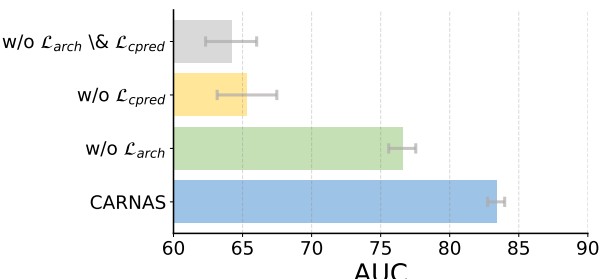

Figure 3: Results of ablation studies on SIDER, where 'w/o $\mathcal{L}_{arch}$' removes $\mathcal{L}_{arch}$ from the overall loss in Eq. (16), 'w/o $\mathcal{L}_{cpred}$' removes $\mathcal{L}_{cpred}$, and 'w/o $\mathcal{L}_{arch}$ & $\mathcal{L}_{cpred}$' removes both of them. The error bars report the standard deviations. Besides, the average and standard deviations of the best-performed baseline on each dataset are denoted as the dark and light thick dash lines respectively.

## E.2 COMPLEXITY ANALYSIS

In this section, we analyze the complexity of our proposed method in terms of its computational time and the quantity of parameters that require optimization. Let's denote by $|V|$ the number of nodes in a graph, by $|E|$ the number of edges, by $|\mathcal{O}|$ the size of search space, and by $d$ the dimension of hidden representations within a traditional graph neural network (GNN) framework. In our approach, $d_0$ represents the dimension of the hidden representations within the identification network $\text{GNN}_0$, $d_1$ represents the dimension of the hidden representations within the shared graph encoder $\text{GNN}_1$, and $d_s$ denotes the dimension within the tailored super-network. Notably, $d_0$ encapsulates the combined dimension of $Q$ chunks, meaning the dimension per chunk is $d_0/Q$.

### E.2.1 TIME COMPLEXITY ANALYSIS

For most message-passing GNNs, the computational time complexity is traditionally $O(|E|d+|V|d^2)$. Following this framework, the $\text{GNN}_0$ in our model exhibits a time complexity of $O(|E|d_0 + |V|d_0^2)$, and the $\text{GNN}_1$ in our model exhibits a time complexity of $O(|E|d_1 + |V|d_1^2)$. The most computationally intensive operation in the invariant architecture customization module, which involves the computation of $\mathcal{L}_{op}$, leads to a time complexity of $O(|\mathcal{O}|^2 d_1)$. The time complexity attributed to the customized super-network is $O(|\mathcal{O}|(|E|d_s + |V|d_s^2))$. Consequently, the aggregate time complexity of our method can be summarized as $O(|E|(d_0 + d_1 + |\mathcal{O}|d_s) + |V|(d_0^2 + d_1^2 + |\mathcal{O}|d_s^2) + |\mathcal{O}|^2 d_1)$.

### E.2.2 PARAMETER COMPLEXITY ANALYSIS

A typical message-passing GNN has a parameter complexity of $O(d^2)$. In our architecture, the disentangled causal subgraph identification network $\text{GNN}_0$ possesses $O(d_0^2)$ parameters, the shared GNN encoder $\text{GNN}_1$ possesses $O(d_1^2)$, the invariant architecture customization module contains $O(|\mathcal{O}|d_1)$ parameters and the customized super-network is characterized by $O(|\mathcal{O}|d_s^2)$ parameters. Therefore, the total parameter complexity in our framework is expressed as $O(d_0^2 + d_1^2 + |\mathcal{O}|d_1 + |\mathcal{O}|d_s^2)$.

The analyses underscore that the proposed method scales linearly with the number of nodes and edges in the graph and maintains a constant number of learnable parameters, aligning it with the efficiency of prior GNN and graph NAS methodologies. Moreover, given that $|\mathcal{O}|$ typically represents a modest constant (for example, $|\mathcal{O}| = 6$ in our search space) and that $d_0$ and $d_1$ is generally much less than $d_s$, the computational and parameter complexities are predominantly influenced by $d_s$. To ensure equitable comparisons with existing GNN baselines, we calibrate $d_s$ within our model such that the parameter count, specifically $|\mathcal{O}|d_s^2$, approximates $d^2$, thereby achieving a balance between efficiency and performance.

### E.3 DYNAMIC TRAINING PROCESS AND CONVERGENCE

For a deeper understanding of our model training process, and further remark the impact of the dynamic $\sigma_p$ in Eq.(17), we conduct experiments and compare the training process in the following settings:

- 'with Dynamic $\sigma$' means we use the dynamic $\sigma_p$ in Eq.(17) to adjust the training key point in each epoch.

- 'w/o Dynamic $\sigma$' means we fix the $\sigma$ in Eq.(16) as a constant value $\frac{\sigma_{max}+\sigma_{min}}{2}$.

According to Figure 4, *our method can converge rapidly in 10 epochs.* Figure 4 also obviously reflects that after 10 epochs the validation loss with dynamic $\sigma$ keeps declining and its *accuracy continuously rising.* However, in the setting without dynamic $\sigma$, the validation loss may rise again, and accuracy cannot continue to improve.

These results verify our aim to adopt this $\sigma_p$ to *elevate the efficiency of model training* in the way of dynamically adjusting the training key point in each epoch by focusing more on the *causal-aware part* (i.e. identifying suitable causal subgraph and learning vectors of operators) in the early stages and focusing more on the performance of the *customized super-network* in the later stages. We also empirically confirm that our method is not complex to train in Appendix E.4.

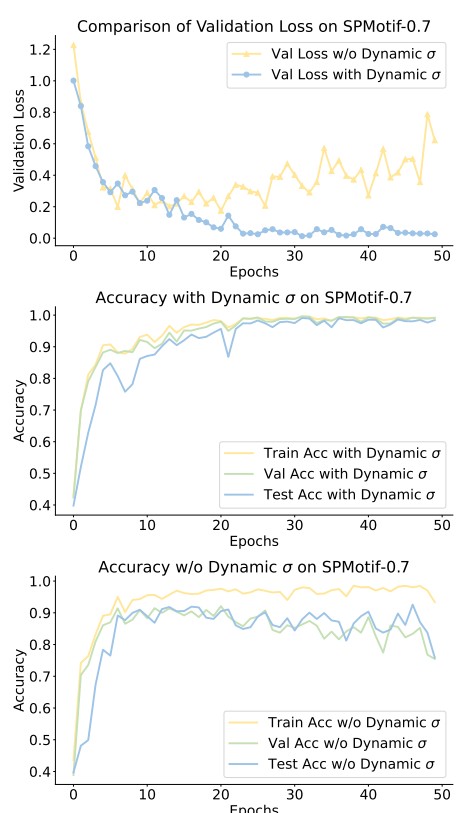

Figure 4: Training process of synthetic datasets.

Furthermore, we report both the training loss and validation loss for the two components ($\mathcal{L}causal$, representing the causal-aware part, and $\mathcal{L}pred$, representing the customized super-network optimization as defined in Equation 16) with and without the dynamic $\sigma$ schedule in Figure 5.

For the training loss, $\mathcal{L}_{pred}$ decreases more steadily and reaches a lower value with less fluctuation under the dynamic schedule. In terms of validation loss, $\mathcal{L}_{pred}$ with the dynamic schedule decreases significantly in later stages, whereas without it, $\mathcal{L}_{pred}$ struggles to converge. Additionally, $\mathcal{L}_{causal}$ without the dynamic schedule exhibits a slight initial increase before decreasing, whereas with the dynamic schedule, it decreases smoothly from the outset. These results indicate that the dynamic

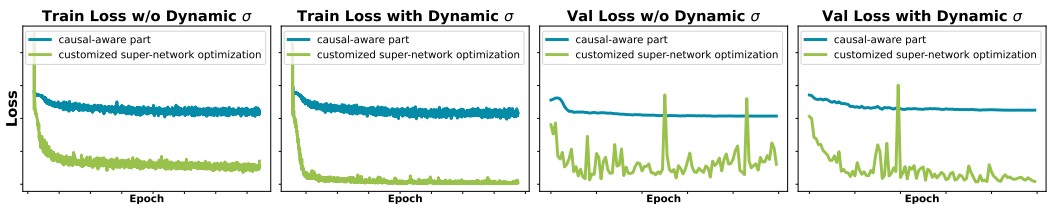

Figure 5: Changes of the two parts of loss.

schedule effectively adjusts the training focus during each epoch. It emphasizes the *causal-aware part* (i.e., identifying suitable causal subgraphs and learning operator vectors) in the early stages and shifts focus to the *customized super-network* performance in later stages.

### E.4 TRAINING EFFICIENCY

To further illustrate the efficiency of CARNAS, we provide a direct comparison with the best-performed NAS baseline, DCGAS, based on the total runtime for 100 epochs. As shown in Table 6, CARNAS consistently requires less time across different datasets while achieving superior best performance, demonstrating its enhanced efficiency and effectiveness.

Table 6: Comparison of runtime

| Method | SPMotif | HIV | BACE | SIDER |
|--------|---------|-----|------|-------|
| DCGAS  | 104 min | 270 min | 12 min | 11 min |
| CARNAS | 76 min  | 220 min | 8 min  | 8 min |

### E.5 HYPER-PARAMETERS SENSITIVITY

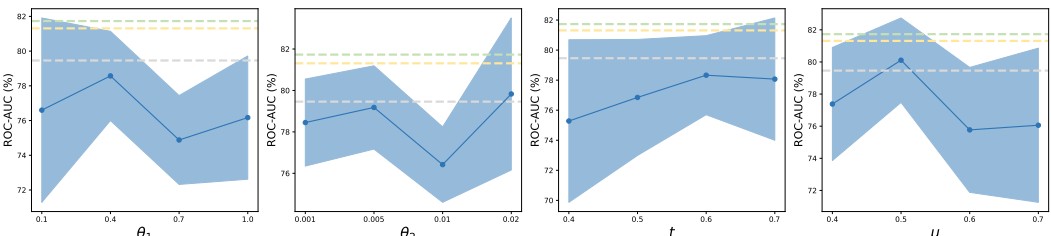

Figure 6: Hyper-parameters sensitivity analysis. The area shows the average ROC-AUC and standard deviations. The green, yellow, grey dashed lines represent the average performance corresponding to the fine-tuned hyper-parameters of CARNAS, best performed baseline DCGAS, 2nd best performed baseline GRACES, respectively.

We empirically observe that our model is insensitive to most hyper-parameters, which remain fixed throughout our experiments. Consequently, the number of parameters requiring tuning in practice is relatively small. $t$, $\mu$, $\theta_1$ and $\theta_2$ have shown more sensitivity, prompting us to focus our tuning efforts on these 4 hyper-parameters.

Therefore, we conduct sensitivity analysis (on BACE) for the 4 important hyper-parameters, as shown in Figure 6. The value selection for these parameters were deliberately varied evenly within a defined range to assess sensitivity thoroughly. The specific hyper-parameter settings used for the CARNAS reported in Table 2 are more finely tuned and demonstrate superior performance to the also finely tuned other baselines. The sensitivity *allows for potential performance improvements* through careful parameter tuning, and our results in sensitivity analysis outperform most baseline methods, indicating a degree of *stability and robustness in response to these hyper-parameters*.

Mention that, the best performance of the fine-tuned DCGAS may exceed the performance of our method without fine-tuning sometimes. This is because, DCGAS addresses the challenge of out-of-distribution generalization through data augmentation, generating a sufficient quantity of graphs for training. In contrast, CARNAS focuses on capturing and utilizing causal and stable subparts to guide

Table 7: Performance Comparison (ROC-AUC) '-' denotes CIGA is not suitable for multi-task dataset

| Class | Method | SIDER | BACE | HIV |
|-------|--------|-------|------|-----|
| Vanilla GNN | GCN | $59.84 \pm 1.54$ | $68.93 \pm 6.95$ | $75.99 \pm 1.19$ |
| | GAT | $57.40 \pm 2.01$ | $75.34 \pm 2.36$ | $76.80 \pm 0.58$ |
| | GIN | $57.57 \pm 1.56$ | $73.46 \pm 5.24$ | $77.07 \pm 1.49$ |
| | SAGE | $56.36 \pm 1.32$ | $74.85 \pm 2.74$ | $75.58 \pm 1.40$ |
| | GraphConv | $56.09 \pm 1.06$ | $78.87 \pm 1.74$ | $74.46 \pm 0.86$ |
| | MLP | $58.16 \pm 1.41$ | $71.60 \pm 2.30$ | $70.88 \pm 0.83$ |
| OOD GNN | ASAP | $55.77 \pm 1.18$ | $71.55 \pm 2.74$ | $73.81 \pm 1.17$ |
| | DIR | $57.34 \pm 0.36$ | $76.03 \pm 2.20$ | $77.05 \pm 0.57$ |
| | MoleOOD | $57.12 \pm 0.82$ | $76.65 \pm 2.71$ | $76.57 \pm 1.11$ |
| | CIGA | - | $77.53 \pm 2.53$ | $76.89 \pm 0.85$ |
| | iMoLD | $60.76 \pm 0.65$ | $78.72 \pm 1.75$ | $77.17 \pm 0.93$ |
| | Coral | $60.32 \pm 1.04$ | $78.65 \pm 1.55$ | $76.88 \pm 1.75$ |
| | DANN | $59.52 \pm 1.02$ | $78.84 \pm 1.11$ | $76.98 \pm 1.32$ |
| | GIL | $59.67 \pm 0.32$ | $75.72 \pm 1.93$ | $73.70 \pm 1.14$ |
| | GSAT | $60.06 \pm 1.11$ | $78.47 \pm 1.70$ | $76.70 \pm 0.98$ |
| | Mixup | $60.83 \pm 0.74$ | $78.16 \pm 2.54$ | $76.81 \pm 1.31$ |
| | GroupDRO | $61.15 \pm 1.06$ | $79.24 \pm 1.30$ | $76.97 \pm 1.36$ |
| | IRM | $59.50 \pm 0.52$ | $78.87 \pm 1.50$ | $76.77 \pm 1.01$ |
| | VREx | $54.60 \pm 0.91$ | $75.77 \pm 3.35$ | $71.60 \pm 1.56$ |
| NAS | DARTS | $60.64 \pm 1.37$ | $76.71 \pm 1.83$ | $74.04 \pm 1.75$ |
| | PAS | $59.31 \pm 1.48$ | $76.59 \pm 1.87$ | $71.19 \pm 2.28$ |
| | GRACES | $61.85 \pm 2.58$ | $79.46 \pm 3.04$ | $77.31 \pm 1.00$ |
| | DCGAS | $63.46 \pm 1.42$ | $81.31 \pm 1.94$ | $78.04 \pm 0.71$ |
| | CARNAS | $\textbf{83.36} \pm \textbf{0.62}$ | $\textbf{81.73} \pm \textbf{2.92}$ | $\textbf{78.33} \pm \textbf{0.64}$ |

the architecture search process. The methodological differences and the resulting disparity in the volume of data used could also contribute to the performance variations observed.

**Limitation.** Although the training time and search efficiency of our method is comparable to most of the Graph NAS methods, we admit that it is less efficient than standard GNNs. At the same time, in order to obtain the best performance for a certain application scenario, our method does need to fine-tune four sensitive hyper-parameters.

# F MORE COMPARISON WITH OOD GNN

In our initial experiment, we compared our model with two non-NAS-based graph OOD methods, ASAP and DIR. We expanded our evaluation to include 13 well-known non-NAS-based graph OOD methods, providing a comprehensive comparison. The results, presented in Table 7, demonstrate that CARNAS not only performs well among NAS-based methods but also significantly outperforms non-NAS graph OOD methods. This superior performance is attributed to CARNAS's ability to effectively discover and leverage the stable causal graph-architecture relationships during the neural architecture search process.

Regarding time and memory costs, Table 8 and 9 show that CARNAS is competitive with non-NAS-based graph OOD methods, as we search the architecture and learn its weights simultaneously. The time and memory efficiency of CARNAS make it a practical choice. Thus, we experimentally verify that the proposed CARNAS does make sense, for addressing the graph OOD problem by diving into the NAS process from causal perspective.

# G CASE STUDY

For graphs with different motif shapes (causal subparts), we present the learned operation probabilities for each layer (in expectation) in Figure 7. The values that are notably higher than others for each layer are highlighted in bold, and the most preferred operators for each layer are listed in the last row.

Table 8: Comparison of Time and Memory Cost between OOD GNN and CARNAS

| Method | SIDER | | BACE | | HIV | |
|---|---|---|---|---|---|---|
| | Time (Mins) | Mem. (MiB) | Time | Mem. | Time | Mem. |
| DIR | 5 | 4328 | 5 | 4323 | 103 | 4769 |
| MoleOOD | 5 | 4317 | 5 | 4315 | 96 | 4650 |
| CIGA | - | - | 4 | 4309 | 86 | 4510 |
| iMoLD | 3 | 4184 | 3 | 4182 | 65 | 4377 |
| Coral | 3 | 4323 | 2 | 4323 | 70 | 4795 |
| DANN | 2 | 4309 | 2 | 4314 | 47 | 4505 |
| GIL | 26 | 4386 | 33 | 4373 | 412 | 6225 |
| GSAT | 4 | 4318 | 4 | 4310 | 49 | 4600 |
| GroupDRO | 4 | 4311 | 10 | 4309 | 50 | 4509 |
| IRM | 4 | 975 | 3 | 978 | 80 | 1301 |
| VREx | 6 | 4313 | 16 | 4314 | 51 | 4516 |
| CARNAS | 8 | 2556 | 8 | 2547 | 220 | 2672 |

Table 9: Performance Comparison on Graph-SST2. All performances are reported under 100 epochs, except for those annotated. Time(Mins), Mem.(MiB).

| Method | Acc | Time | Mem. | Method | Acc | Time | Mem. |
|---|---|---|---|---|---|---|---|
| Coral | $77.28 \pm 1.98$ | 62 | 4820 | DANN | $77.96 \pm 3.50$ | 38 | 4679 |
| DIR | $67.90 \pm 10.08$ | 171 | 4891 | DIR (200 epochs) | $79.19 \pm 1.85$ | 326 | 4891 |
| GIL | $75.53 \pm 6.01$ | 418 | 5628 | GIL (200 epochs) | $78.67 \pm 1.48$ | 816 | 5629 |
| GSAT | $78.79 \pm 1.85$ | 36 | 4734 | Mixup | $78.76 \pm 2.00$ | 31 | 4682 |
| ERM | $75.99 \pm 3.25$ | 29 | 4667 | GroupDRO | $76.97 \pm 3.49$ | 28 | 4695 |
| IRM | $78.12 \pm 1.73$ | 70 | 1389 | VREx | $79.62 \pm 1.26$ | 27 | 4692 |
| CIGA | $65.62 \pm 7.87$ | 157 | 4683 | CIGA (200 epochs) | $79.98 \pm 1.61$ | 306 | 4683 |
| CARNAS | $80.58 \pm 1.72$ | 199 | 2736 | | | | |

We observe that different motif shapes indeed prefer different architectures, e.g., graphs with cycle prefer GAT in the third layer, while this operator is seldomly chosen in neither layer of the other two types of graphs; the operator distributions are similar for graphs with cycle and house in the first layer, but differ in other layers. To be specific, Motif-Cycle is characterized by a closed-loop structure where each node is connected to two neighbors, displaying both symmetry and periodicity. For graphs with this motif, CARNAS identifies SAGE-GCN-GAT as the most suitable architecture. Motif-House, on the other hand, features a combination of triangular and quadrilateral structures, introducing a certain level of hierarchy and asymmetry. For graphs with this shape, CARNAS determines that GIN-MLP-GCN is the optimal configuration. Lastly, Motif-Crane presents more complex cross-connections between nodes compared to the previous two motifs, and CARNAS optimally configures graphs with it with a GIN-SAGE-GCN architecture.

By effectively integrating various operations and customizing specific architectures for different causal subparts (motifs) with diverse features, our NAS-based CARNAS can further improve the OOD generalization.

To better illustrate the learned graph-architecture relationship, we also visualize the causal subgraphs for each dataset in our case study in Figure 8.

# H   RELATED WORK

## H.1   GRAPH NEURAL ARCHITECTURE SEARCH

In the rapidly evolving domain of automatic machine learning, Neural Architecture Search (NAS) represents a groundbreaking shift towards automating the discovery of optimal neural network architectures. This shift is significant, moving away from the traditional approach that heavily relies on manual expertise to craft models. NAS stands out by its capacity to autonomously identify archi-

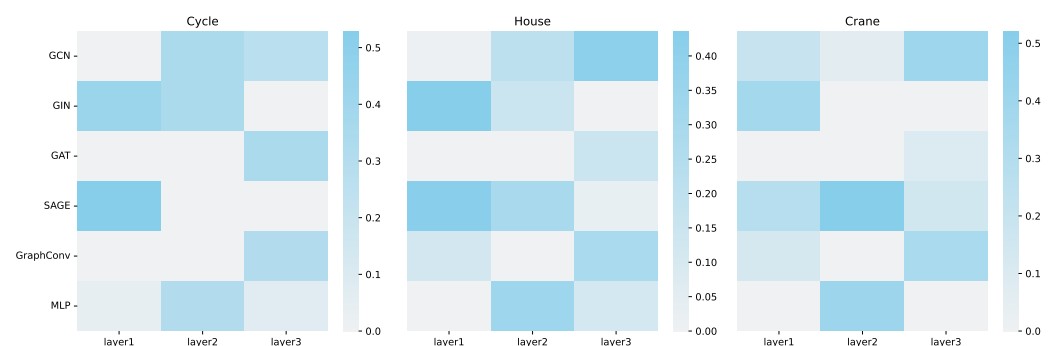

Figure 7: Comparison of operation probabilities for graphs with different motif shapes.

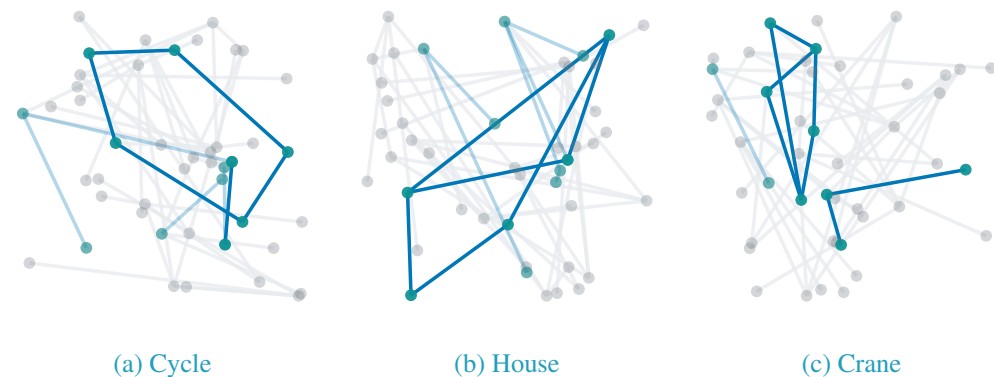

(a) Cycle        (b) House        (c) Crane

Figure 8: Visualization of edge importance for forming causal subgraphs in SP-Motif Dataset. Structures with deeper colors mean higher importance.

tectures that are finely tuned for specific tasks, demonstrating superior performance over manually engineered counterparts. The exploration of NAS has led to the development of diverse strategies, including reinforcement learning (RL)-based approaches (76; 18), evolutionary algorithms-based techniques (43; 32), and methods that leverage gradient information (31; 62). Among these, graph neural architecture search has garnered considerable attention.

The pioneering work of GraphNAS (11) introduced the use of RL for navigating the search space of graph neural network (GNN) architectures, incorporating successful designs from the GNN literature such as GCN, GAT, etc. This initiative has sparked a wave of research (11; 51; 41; 4; 13; 74; 12), leading to the discovery of innovative and effective architectures. Recent years have seen a broadening of focus within Graph NAS towards tackling graph classification tasks, which are particularly relevant for datasets comprised of graphs, such as those found in protein molecule studies. This research area has been enriched by investigations into graph classification on datasets that are either independently identically distributed (51) or non-independently identically distributed, with GRACES (41) and DCGAS (61) being notable examples of the latter. Through these efforts, the field of NAS continues to expand its impact, offering tailored solutions across a wide range of applications and datasets.

## H.2 GRAPH OUT-OF-DISTRIBUTION GENERALIZATION

In the realm of machine learning, a pervasive assumption posits the existence of identical distributions between training and testing data. However, real-world scenarios frequently challenge this assumption with inevitable shifts in distribution, presenting significant hurdles to model performance in out-of-distribution (OOD) scenarios (46; 70; 71). The drastic deterioration in performance becomes evident when models lack robust OOD generalization capabilities, a concern particularly pertinent in the domain of Graph Neural Networks (GNNs), which have gained prominence within the graph community (27). Several noteworthy studies (55; 54; 28; 9; 47; 30; 48) have tackled this

challenge by focusing on identifying environment-invariant subgraphs to mitigate distribution shifts. These approaches typically rely on pre-defined or dynamically generated environment labels from various training scenarios to discern variant information and facilitate the learning of invariant subgraphs. (52; 68) have divided recent literature that solve the graph OOD generalization problem, into three categories: 1) Graph augmentation methods (75; 45; 10; 64) enhance OOD generalization by increasing the quantity and diversity of training data through systematic graph modifications. 2) The second type of methods (34; 26) develop new graph models to learn OOD-generalized representations. 3)The third type of methods (69; 17) enhance OOD generalization through tailored training schemes with specific objectives and constraints. There are various datasets and benchmarks (16; 36; 14; 20; 21) help for assessing generalizability and adaptability. Moreover, the existing methods usually adopt a fixed GNN encoder in the whole optimization process, neglecting the role of graph architectures in out-of-distribution generalization. In this paper, we focus on automating the design of generalized graph architectures by discovering causal relationships between graphs and architectures, and thus handle distribution shifts on graphs.

## H.3    CAUSAL LEARNING ON GRAPHS

The field of causal learning investigates the intricate connections between variables (40; 38), offering profound insights that have significantly enhanced deep learning methodologies. Leveraging causal relationships, numerous techniques have made remarkable strides across diverse computer vision applications (65; 35). Additionally, recent research has delved into the realm of graphs (72; 73). For instance, (57) implements interventions on non-causal components to generate representations, facilitating the discovery of underlying graph rationales. (9) decomposes graphs into causal and bias subgraphs, mitigating dataset biases. (29) introduces invariance into self-supervised learning, preserving stable semantic information. (6) ensures out-of-distribution generalization by capturing graph invariance. (19) tackled the challenge of learning causal graphs involving latent variables, which are derived from a mixture of observational and interventional distributions with unknown interventional objectives. To mitigate this issue, the study proposed an approach leveraging a $\Psi$-Markov property. (1) introduced a randomized algorithm, featuring $p$-colliders, for recovering the complete causal graph while minimizing intervention costs. Additionally, (3) presented an adaptable method for causality detection, which notably benefits from various types of interventional data and incorporates sophisticated neural architectures such as normalizing flows, operating under continuous constraints. However, these methods adopt a fixed GNN architecture in the optimization process, neglecting the role of architectures in causal learning on graphs. In contrast, in this paper, we focus on handling distribution shifts in the graph architecture search process from the causal perspective by discovering the causal relationship between graphs and architectures. There are also works (50; 23; 66; 67) employ GNN for causality learning.

