# OpenReview forum: "Causal-aware Graph Neural Architecture Search under Distribution Shifts"
_ICLR.cc/2025/Conference — Submitted to ICLR 2025_

### Official Review · Reviewer_Q7m9 · 2024-10-24

**Soundness:** 3
**Presentation:** 3
**Contribution:** 2
**Rating:** 5
**Confidence:** 4

**Summary:**

This paper presents CARNAS (Causal-aware Graph Neural Architecture Search), a novel approach that addresses the challenge of distribution shifts in Graph Neural Architecture Search.
Unlike existing methods that rely on potentially spurious correlations between graphs and architectures, CARNAS focuses on discovering and leveraging causal relationships to achieve better generalization.

The solution consists of three main components:

1. Disentangled Causal Subgraph Identification: Discovers subgraphs with stable predictive capabilities across different distributions
2. Graph Embedding Intervention: Works in latent space to preserve essential predictive features while removing non-causal elements
3. Invariant Architecture Customization: Reinforces causal invariance and uses it to design generalized architectures

The approach's effectiveness is validated through experiments on both synthetic and real-world datasets, demonstrating superior out-of-distribution generalization compared to existing methods.

**Strengths:**

1. The application of neural architecture search to address graph out-of-distribution (OOD) problems represents a significant innovation compared to traditional fixed-backbone approaches, as demonstrated in Appendix G.3. This shift from static to adaptive architectures opens new possibilities for graph OOD generalization.
2. The method's effectiveness is convincingly demonstrated through comprehensive experiments on SPMotif and OGBG-Mol* datasets, where it consistently outperforms existing approaches in handling distribution shifts.
3. The paper stands out for its clear and organized presentation, featuring well-designed figures that effectively illustrate complex concepts, complemented by rigorous mathematical formulations that provide a solid theoretical foundation.

**Weaknesses:**

1. Computational Efficiency Concerns:  Section 3.3 reveals a potential limitation. The method searches through an extensive space of graph neural networks across all layers, which could be computationally more intensive than existing baselines. This increased computational and memory overhead might present challenges when applied to large-scale graphs.
2. Limited Experimental Validation: While DIR is included as a baseline, other important causal subgraph-based methods from recent works (such as [1],[2]) are not considered.
Additionally, the evaluation datasets - SPMotif (synthetic) and OGBG-Mol* (relatively small graph sizes) - leave questions about scalability.
It would be valuable to see performance on larger-scale datasets like DrugOOD or GOOD to address concerns about memory consumption and computational time.
3. Novelty Discussion:
While the neural architecture search component is innovative, the underlying methodology shares significant similarities with existing graph OOD approaches like DIR and related works [1-3]. The core mechanisms - using weighted top-k for causal subgraph identification and random combination for spurious subgraph intervention - closely parallel previous methods. This raises questions about the method's novelty beyond the architecture search component.
[1] Learning causally invariant representations for out-of-distribution generalization on graphs.
[2]Learning invariant graph representations for out-of-distribution generalization.
[3] Improving subgraph recognition with variational graph information bottleneck

**Questions:**

1. Could you conduct an ablation study focusing on the neural network search details provided in the added Appendix C.1?
Specifically, I'm interested in understanding:
- The effectiveness of different backbones to OOD distribution shifts, possibly illustrated through weight distributions.
- How about time and memory requirements during the search process?
- Are all of the backbones crucial for effective OOD distribution handling?

2. How does your method behave in the large graph compared to the previous fixed-network methods since you may search in a large network space?  Will it be out of memory or the time computation will exponentially grow？

---

> ### Author Response · Authors · 2024-11-26
>
> We are grateful to the reviewer for the insightful comments and suggestions. We have meticulously reviewed each point and provide the following responses:
>
> > W1. Computational Efficiency Concerns: Section 3.3 reveals a potential limitation.
> >
> >
> > W2-2. Evaluation datasets: Performance on DrugOOD or GOOD to address concerns about memory consumption and computational time.
> >
> > Q2. How does your method behave in the large graph compared to the previous fixed-network methods since you may search in a large network space? Will it be out of memory or the time computation will exponentially grow？
> >
>
> A1. Thank you for your valuable feedback. Since DrugOOD is similar to OGBG-Mol* (both involving molecule datasets with comparable graph sizes), we conducted additional experiments on the GOODSST2 dataset from GOOD. This dataset, converted from text sequences, presents a different domain compared to molecule datasets. Here, nodes represent words, edges indicate relations between words, and labels reflect sentence sentiment.
>
> We compared performance, time, and memory costs across methods. To ensure fairness, we recorded results after 100 epochs for all methods. Additionally, since methods like DIR, GIL, and CIGA did not achieve optimal performance within 100 epochs, we also recorded their results after 200 epochs.
>
> | **Method** | **Acc** | **Time (Mins)** | **Mem. (MiB)** | **Method** | **Acc** | **Time (Mins)** | **Mem. (MiB)** |
> | --- | --- | --- | --- | --- | --- | --- | --- |
> | Coral | 77.28 ± 1.98 | 62 | 4820 | DANN | 77.96 ± 3.50 | 38 | 4679 |
> | DIR | 67.90 ± 10.08 | 171 | 4891 | DIR (200 epochs) | 79.19 ± 1.85 | 326 | 4891 |
> | GIL | 75.53 ± 6.01 | 418 | 5628 | GIL (200 epochs) | 78.67 ± 1.48 | 816 | 5629 |
> | GSAT | 78.79 ± 1.85 | 36 | 4734 | Mixup | 78.76 ± 2.00 | 31 | 4682 |
> | ERM | 75.99 ± 3.25 | 29 | 4667 | GroupDRO | 76.97 ± 3.49 | 28 | 4695 |
> | IRM | 78.12 ± 1.73 | 70 | 1389 | VREx | 79.62 ± 1.26 | 27 | 4692 |
> | CIGA | 65.62 ± 7.87 | 157 | 4683 | CIGA (200 epochs) | 79.98 ± 1.61 | 306 | 4683 |
> | CARNAS | 80.58 ± 1.72 | 199 | 2736 |  |  |  |  |
>
> **CARNAS achieves the best performance while maintaining competitive time and memory costs compared to non-NAS-based (fixed-network) graph OOD methods.** The efficiency of the architecture search process stems from CARNAS's ability to **simultaneously search for the architecture and optimize its parameters, minimizing additional computational overhead**. This demonstrates the practical viability and efficiency of CARNAS, even for larger-scale graph datasets.
>
> *Following your advice, we have included these new results on the GOODSST2 dataset in `Appendix F` in the revised [**`pdf`**](https://openreview.net/pdf?id=58AhfT4Zz1). Additionally, we have cited both DrugOOD and GOOD for their valuable contributions to graph benchmarks.*

---

> > ### Author Response · Authors · 2024-11-26
> >
> > > W2-1. *"Limited Experimental Validation: While DIR is included as a baseline, other important causal subgraph-based methods from recent works (such as [1],[2]) are not considered."*
> > >
> >
> > A2. Thank you for your comment. In our main paper, we compared our model with two non-NAS-based graph OOD methods, ASAP and DIR. However, we have also expanded our evaluation to include **13 well-known non-NAS-based graph OOD methods**, *encompassing all the methods you mentioned ([1] as CIGA and [2] as GIL)*.
> >
> > The results, provided in `Table 7 & 9`, demonstrate that **CARNAS not only performs exceptionally well among NAS-based methods but also significantly outperforms non-NAS-based graph OOD methods.** This improvement highlights the effectiveness of CARNAS in discovering and leveraging stable causal graph-architecture relationships during the neural architecture search process.
> >
> > > W3. Novelty discussion beyond the architecture search component.
> > >
> >
> > A3. Thank you for your comment.
> >
> > We would like to emphasize that the main novelty and contribution of our work lies in **unveiling and utilizing the *causal graph-architecture relationship* to address distribution shifts during the graph neural architecture search (NAS) process.** This is an **under-explored yet crucial issue**, as previous works ([1-3]) have highlighted the complex relationship between graph data and optimal architectures. They suggest that *different subparts of a graph instance with varying features may be suited to different architectures.*
> >
> > On this basis, we **first propose to jointly optimize causal graph-architecture relationship and architecture search** by offering an **end-to-end** training strategy for extracting and utilizing causal relationships between graph data and architecture, which is stable under distribution shifts, during the architecture search process, thereby enhancing the model’s capability of OOD generalization.
> >
> > We further clarify the differences from existing methods:
> >
> > - In Disentangled Causal Subgraph Identification module, we introduce the ideology of *disentangled learning*[4,5] to delineates distinct latent factors as unique vector representations, thus comprehensively unveil and disentangle causal latent graph structural features from spurious features that influence constructing architecture. Together with following two modules, it ***disentangles complex graph-architecture relationships*** by learning distinct latent factors, influencing the customization of optimal architecture, from complicated graph structure features. Thereby, it enables a more precise extraction of causal subgraph which is leveraged to derive the optimal architecture.
> > - In Graph Embedding Intervention module, we perform interventions on causal subgraph $G_c$ with non-causal subgraphs in latent space as Eq.10 to obtain intervention graphs, that is essential for the next Invariant Architecture Customization module. We customize the optimal architecture $A_c$ with $G_c$ and other simulated intervention architectures with intervention graphs, using $\mathcal{L}_{arch}$ for simulated intervention architectures to regulate the influence of spurious parts in constructing the optimal architecture. This approach helps form a stable architecture solely based on the causal subgraph. The ***intervention on not just graph but especially further on architecture, makes our method significantly different from previous work.***
> >
> > To summarize, our approach **differs in both objective and methodology.**
> >
> > ---
> >
> > [1]Design space for graph neural networks. NeurIPS20
> >
> > [2]Principal neighbourhood aggregation for graph nets. NeurIPS20
> >
> > [3]How neural networks extrapolate: From feedforward to graph neural networks. arXiv20
> >
> > [4]Representation learning: A review and new perspectives.
> >
> > [5]Disentangled representation learning.

---

> > > ### Author Response · Authors · 2024-11-26
> > >
> > > > Q1. Could you conduct an ablation study focusing on the neural network search details provided in the added Appendix C.1? Specifically, I'm interested in understanding:
> > > >
> > > > - The effectiveness of different backbones to OOD distribution shifts, possibly illustrated through weight distributions.
> > > > - How about time and memory requirements during the search process?
> > > > - Are all of the backbones crucial for effective OOD distribution handling?
> > >
> > > A4. Thank you for your question. As shown in `Tables 1, 2, and 7`, we compare the performance of CARNAS with each of the six backbones—GCN, GAT, GIN, SAGE, GraphConv, and MLP—that we adopted as candidate operations, as described in `Appendix D.1`. Our method consistently outperforms these backbones individually, which serves as an ablation study demonstrating the effectiveness of integrating backbones through the neural architecture search (NAS) process. Below, we address your specific questions in detail:
> > >
> > > - A4-1. Regarding the effectiveness of different backbones/operations under distribution shifts: The case study in `Appendix G` addresses this issue. For graphs with different motif structures (causal subcomponents), we provide the learned operation probabilities for each layer (in expectation) in `Figure 7`. These probabilities, *i.e., weight distributions*, reveal that different graph types (characterized by diverse motif shapes) prefer distinct architectures. Notably, each of the six backbones is selected as the most suitable operation for at least one graph’s one layer, emphasizing their necessity. Furthermore, our approach is flexible and can accommodate additional backbones for specific tasks or datasets. By integrating diverse operations and tailoring architectures to match graphs with varying causal subparts and features, our NAS-based CARNAS significantly enhances OOD generalization.
> > > - A4-2. Regarding time and memory requirements during the search process: The architecture search module is jointly optimized alongside other modules, so we report the statistics for the whole process. As reported in `Table 8 & 9` and above rebuttal answer `A1.`, the **overall time and memory costs for CARNAS are competitive with those of non-NAS-based (fixed-network) graph OOD methods**. This efficiency is achieved by simultaneously searching for the architecture and learning its parameters. The practical time and memory efficiency of CARNAS makes it a viable and effective choice. Thus, we experimentally verify that the proposed CARNAS addresses distribution shifts during the NAS process from a causal perspective.
> > > - A4-3. Concerning the necessity of all six backbones for effective OOD handling: As illustrated in `Figure 7`, none of the six backbones exhibits a zero probability of being selected. This indicates that each backbone has the potential to be preferred in certain contexts, making them indispensable as candidate operations in OOD scenarios. Their diverse contributions are crucial for achieving the strong generalization capabilities of CARNAS on OOD datasets.
> > >
> > > We extend our sincere thanks again for your invaluable feedback and thoughtful consideration.

---

### Official Review · Reviewer_EUwC · 2024-10-24

**Soundness:** 1
**Presentation:** 3
**Contribution:** 2
**Rating:** 3
**Confidence:** 5

**Summary:**

The study proposes a novel method called Causal-aware Graph Neural Architecture Search (CARNAS) to address the challenges posed by distribution shifts in the process of Graph Neural Architecture Search (Graph NAS). Existing methods face limitations when handling distribution shifts in real-world scenarios, as the correlations they exploit between graphs and architectures are often spurious and subject to variation across different distributions. CARNAS aims to discover and leverage the causal relationship between graphs and architectures to search for optimal architectures capable of maintaining generalization under distribution shifts.

**Strengths:**

1. This paper innovatively proposes using NAS to address the problem of causal information identification in graph data.
2. The paper conducts extensive experiments to validate the proposed method.

**Weaknesses:**

1. I believe the paper does not clearly explain why NAS can help adjust GNNs to identify causal information, which I consider the main issue of the paper. In my view, NAS optimizes the structure of GNNs, enhancing their efficiency or expressiveness, but it does not inherently enable GNNs to determine what type of data to model. At the very least, the authors did not provide a clear explanation of this point in the paper.
2. The paper lacks theoretical justification for the regulatory capability of NAS.
3. The survey and introduction of related work are insufficient.

**Questions:**

Could you explain how NAS can guide GNNs to model specific data causal relationships?

---

> ### Author Response · Authors · 2024-11-26
>
> We would like to express our sincere appreciation to the reviewer for providing us with detailed comments and suggestions. We have carefully reviewed each point and respond to the reviewer’s comments point by point as follows.
>
> > W1. *“I believe the paper does not clearly explain why NAS can help adjust GNNs to identify causal information, which I consider the main issue of the paper. In my view, NAS optimizes the structure of GNNs, enhancing their efficiency or expressiveness, but it does not inherently enable GNNs to determine what type of data to model. At the very least, the authors did not provide a clear explanation of this point in the paper.”*
> >
> >
> > Q1. Could you explain how NAS can guide GNNs to model specific data causal relationships?
> >
>
> A1. Thank you for your question. We believe there is a misunderstanding regarding the target problem of this work. We would like to clarify that this work utilizes causality to enhance NAS rather than employing NAS to identify causal information. We apologize for the confusion and will further clarify this in the revised manuscript.
>
> To be specific, we frame the entire predicting pipeline in graph task as graph $\rightarrow$ architecture $\rightarrow$ label. Our work aims to **unveil and utilize the *causal graph-architecture relationship* to address distribution shifts during the graph neural architecture search (NAS) process**. This is an **under-explored yet important issue**, as previous works [1-3] emphasize the complex relationship between graph data and the optimal architecture. However, prior methods fix the network structure and solely focus on the *graph-label relationship*, neglecting the influence of the architecture itself, which can lead to suboptimal results under distribution shifts.
>
> As you mentioned, NAS optimizes the structure of GNNs (based on graph data), and this process provides an opportunity to delve into the graph-architecture relationship. Therefore, we incorporate the architecture search process to improve generalization by identifying and leveraging the causal graph-architecture relationship to construct the optimal architecture (denoted as $A_c$ in our paper) for graph data.
>
> Specifically, in our methods:
>
> 1. The causal subgraph $G_c$, extracted from the *Disentangled Causal Subgraph Identification module*, serves as a carrier of information causally relevant to the optimal architecture, rather than being merely causally related to the label $\hat{Y}$ as in prior approaches.
> 2. In the *Invariant Architecture Customization module*, we customize the optimal architecture $A_c$ using $G_c$ and simulated intervention architectures ${A_v}_j$s using intervention graphs. The loss term  $\mathcal{L}\_{arch}$ for ${A_v}_j$s is used to **regulate the influence of spurious components on the construction of the optimal architecture**. This allows us to identify and leverage causal graph-architecture relationships to build the optimal architecture.
>
> In conclusion, we first propose to **jointly optimize causal graph-architecture relationship and architecture search** by offering an **end-to-end** training strategy for extracting and utilizing causal relationships between graph data and architecture, which is stable under distribution shifts, during the architecture search process, thereby enhancing the model’s capability of OOD generalization.
>
> > W2. The paper lacks theoretical justification for the regulatory capability of NAS.
> >
>
> A2. Thank you for your comment. While our work primarily focuses on the empirical effectiveness of Neural Architecture Search (NAS), it is worth noting that NAS has been widely studied[4-8] for its ability to regulate and optimize neural network performance across diverse tasks.
>
> Moreover, we have included additional **theoretical analysis about the problem to further illustrate our method in a more rigorous way**. The added theoretical analysis is highlighted in blue in `Appendix C` in the **updated [`pdf`](https://openreview.net/pdf?id=58AhfT4Zz1)**.
>
> ---
>
> [1] You et al. Design space for graph neural networks. NeurIPS2020
>
> [2] Corso et al. Principal neighbourhood aggregation for graph nets. NeurIPS2020
>
> [3] Xu et al. How neural networks extrapolate: From feedforward to graph neural networks. arXiv2020
>
> [4] DARTS: Differentiable Architecture Search. *ICLR 19.*
>
> [5] Auto-GNN: Neural Architecture Search of Graph Neural Networks. *2019*.
>
> [6] Graph Neural Architecture Search. *IJCAI 20*.
>
> [7] Automated Machine Learning on Graphs: A Survey. *IJCAI 21*.
>
> [8] One-shot Graph Neural Architecture Search with Dynamic Search Space. *AAAI 21*.

---

> > ### Author Response · Authors · 2024-11-26
> >
> > > W3. *"The survey and introduction of related work are insufficient."*
> > >
> >
> > A3. Thank you for your comment. In our original paper, due to the page limitations of the main paper, we included a concise introduction of related work in `Section 5` and provided **additional review of related work in `Appendix H`**. Specifically:
> >
> > - `Appendix H.1` introduces the development of **Graph Neural Architecture Search** (GraphNAS), including foundational and recent research efforts focused on reinforcement learning and other strategies to optimize GNN architectures for various tasks, such as graph classification.
> > - `Appendix H.2` reviews advancements in **out-of-distribution generalization for GNNs**, mentioning existing methods that focus on identifying environment-invariant subgraphs to mitigate distribution shifts. It also explains our novel approach of leveraging causal relationships between graphs and architectures for robust generalization during NAS process.
> > - `Appendix H.3` discusses **causal learning on graphs**, summarizing existing methods that incorporate causality for improved representation and generalization, and highlights our focus on automating architecture design by discovering causal relationships between graphs and GNN architectures to address distribution shifts in NAS process.
> >
> > *Following your suggestion, we have further added more references and expanded the discussion of related works in `Appendix H` to provide a more comprehensive survey. If you believe there are still missing references, please feel free to let us know, and we are happy to include them.*

---

> > ### Comment · Reviewer_EUwC · 2024-11-26
> > **Thanks**
> >
> > Thanks for the answers, yet it still confuses me how enhancing NAS helps in handling distribution shifts

---

> > > ### Author Response · Authors · 2024-12-04
> > >
> > > > How enhancing NAS helps in handling distribution shifts.
> > > >
> > >
> > > A4. Thank you for your follow-up response. We would like to clarify this in two points based on `A1`:
> > >
> > > 1. As already discussed in `A1`, the **primary goal of this work** is to address **distribution shifts during the graph neural architecture search (NAS) process *itself*.** Tackling the “distribution shifts problem in the NAS process *itself*” is a **critical yet under-explored challenge**, as prior studies [1-3] underscore the intricate relationship between graph data and the optimal architecture.
> > > 2. Furthermore, solving the problem of “NAS under distribution shifts,” as outlined above, also directly **contributes to improving overall prediction generalization**. As clarified in `A1`, we conceptualize the prediction pipeline in graph tasks as graph $\rightarrow$ architecture $\rightarrow$ label. Existing methods typically fix the network structure and focus only on the *graph-label relationship*, overlooking the influence of the architecture itself. This oversight can lead to suboptimal results under distribution shifts. In contrast, our approach *dynamically constructs the optimal architecture for each graph instance by mitigating the effects of spurious graph-architecture relationships during the NAS process*. By **leveraging a customized causal optimal architecture tailored to each input graph instance**, our method achieves superior prediction performance under distribution shift scenarios.
> > >
> > > We hope this more detailed explanation clarifies your confusion.

---

### Official Review · Reviewer_Zb28 · 2024-11-03

**Soundness:** 3
**Presentation:** 2
**Contribution:** 3
**Rating:** 6
**Confidence:** 3

**Summary:**

This paper addresses the challenge of graph neural architecture search (Graph NAS) under distribution shifts. The authors observe that existing Graph NAS methods fail to generalize well when there are distribution shifts between training and testing data, since they may exploit spurious correlations that don't hold across distributions. To tackle this, they propose CARNAS (Causal-aware Graph Neural Architecture Search), a novel approach that discovers and leverages causal relationships between graphs and architectures.

The main contributions of this work is 1) it is the first work to study Graph NAS under distribution shifts from a causal perspective; 2) they propose a novel framework with a disentangled Causal Subgraph Identification to find stable predictive subgraphs, a Graph Embedding Intervention component to validate causality in latent space and Invariant Architecture Customization to handle distribution shifts. Comprehensive experiments on synthetic and real-world datasets showing superior out-of-distribution generalization compared to existing methods.

**Strengths:**

1. The paper is the first to study Graph NAS under distribution shifts using causality. The problem is well-motivated with clear real-world relevance and applications.
2. The authors present comprehensive experiments on both synthetic and real-world datasets that demonstrate clear performance improvements over existing baselines. The thorough ablation studies effectively validate each component of the proposed method, and the analysis provides valuable insights into the model's behavior.
3. The paper is well-structured. The experimental analysis is clearly presented, making the work reproducible.

**Weaknesses:**

1. While the paper shows good performance on the tested datasets, it lacks a detailed analysis of computational complexity and memory requirements. Specifically, the time complexity of $O(|E|(d_0 + d_1 + |O|d_s) + |V|(d_0^2 + d_1^2 + |O|d_s^2) + |O|^2d_1)$ could become prohibitive for very large graphs. The authors should discuss how their method performs on graphs with millions of nodes and edges, which are common in real-world applications like social networks.
2. The method requires careful tuning of four critical hyperparameters ($t$, $\mu$, $\theta_1$, $\theta_2$), which may significantly impact performance. In particular, the edge importance threshold t in Eq.(6) and the intervention intensity $\mu$ in Eq.(10) show high sensitivity in experiments. While the authors provide some sensitivity analysis on BACE dataset, they don't fully explain how to effectively tune these parameters for new datasets or application domains.
3. The paper lacks formal theoretical guarantees for the causal discovery process. While the empirical results are strong, the authors should clarify under what conditions their method is guaranteed to identify true causal relationships and provide bounds on the probability of discovering spurious correlations. Additionally, the relationship between the intervention loss and causal invariance could be more rigorously established.

**Questions:**

1. In the graph embedding intervention module, you use $\mu$ to control intervention intensity in Eq.(10): $H_{v_j} = (1-μ)·H_c + μ·H_{s_j}$. Have you considered using adaptive intervention strategies where $\mu$ varies based on the structural properties of $G_c$ and $G_s$? This could potentially better handle graphs with varying degrees of spurious correlations.
2. The overall objective function (Eq.(17)) uses a linearly growing $\sigma_p$ corresponding to epoch number. Could you elaborate on why linear growth was chosen over other schedules (e.g., exponential, step-wise)? How does the schedule of $\sigma_p$ affect the trade-off between causal structure learning and architecture optimization?

---

> ### Author Response · Authors · 2024-11-26
>
> We would like to express our sincere gratitude to the reviewer for providing us with detailed comments and insightful questions. We have carefully considered the reviewer's feedback and would like to address each point as follows:
>
> > W1. While the paper shows good performance on the tested datasets, it lacks a detailed analysis of computational complexity and memory requirements. Specifically, the time complexity could become prohibitive for very large graphs. The authors should discuss how their method performs on graphs with millions of nodes and edges, which are common in real-world applications like social networks.
> >
>
> A1. Thank you for your comment. We provide a detailed analysis of computational complexity and memory requirements in `Appendix E.2, E.4, and F`. Specifically, as shown in `Table 6`, CARNAS consistently requires less time while achieving superior performance compared to the best-performing NAS baseline, demonstrating its enhanced efficiency and effectiveness. Additionally, `Table 8 & 9` shows that **CARNAS's overall time and memory costs are competitive with non-NAS-based (fixed-network) graph OOD methods**, achieved by **simultaneously searching for architectures and learning parameters**. The practical time and memory efficiency of CARNAS makes it a viable and effective choice. We would also like to clarify that our work, like many previous Graph OOD studies [1,2], focuses on the task of graph classification, where the size of each graph instance does not typically contain millions of nodes and edges. Graphs with millions of nodes and edges, as you mentioned, are usually used for node classification (or link prediction). The task of node classification can be viewed as graph classification based on the ego-graphs of a node, where the K-hop neighbor graph of each node serves as a graph instance that are significantly smaller than the original whole graph, allowing our model to generalize to node classification with comparable complexity to traditional methods.
>
> > W2. The method requires careful tuning of four critical hyperparameters, which may significantly impact performance. In particular, the edge importance threshold $t$ in Eq.(6) and the intervention intensity $\mu$ in Eq.(10) show high sensitivity in experiments. While the authors provide some sensitivity analysis on BACE dataset, they don't fully explain how to effectively tune these parameters for new datasets or application domains.
> >
>
> A2. Thank you for your comment. The edge importance threshold $t$ is used to predefine the ratio of extracting causal subgraphs from original graph instance and $\mu$ is used to predefine the intervention intensity. The values of these two hyper-parameters can be roughly determined based on domain knowledge relevant to the dataset. In practical, we tune the parameters by giving a recommended range for each of them, and conduct hyper-parameter optimization on validation set. The range for SPMotif is  $t\in [0.5,0.85], \mu\in[0.2,0.5], \theta_1\in[0.3,0.5], \theta_2\in[0.005,0.015]$ and for OGBG-Mol* is $t\in [0.4,0.7], \mu\in[0.4,0.7], \theta_1\in[0.1,1.0], \theta_2\in[0.001,0.020]$.
>
> > W3. The paper lacks formal theoretical guarantees for the causal discovery process. While the empirical results are strong, the authors should clarify under what conditions their method is guaranteed to identify true causal relationships and provide bounds on the probability of discovering spurious correlations. Additionally, the relationship between the intervention loss and causal invariance could be more rigorously established.
> >
>
> A3. Thank you for your invaluable advice! Although the rebuttal time is limited, we have tried to provide a detailed theoretical analysis to further illustrate our method in a more rigorous way. **The added theoretical analysis is highlighted in blue in `Appendix C` in the updated [`pdf`](https://openreview.net/pdf?id=58AhfT4Zz1)**. It step by step introduces how we propose to **jointly optimize the causal graph-architecture relationship and architecture search** by offering an **end-to-end** training strategy. This strategy enables the extraction and utilization of causal relationships between graph data and architecture, which remain stable under distribution shifts during the architecture search process, thereby enhancing the model’s capability for OOD generalization.

---

> > ### Author Response · Authors · 2024-11-26
> >
> > > Q1. In the graph embedding intervention module, you use $\mu$ to control intervention intensity in Eq.(10): $\mathbf{H_v}_j = (1-\mu)\cdot{\mathbf{H_c}} + \mu\cdot{\mathbf{H_s}}_j$. Have you considered using adaptive intervention strategies where $\mu$ varies based on the structural properties of $G_c$ and $G_s$? This could potentially better handle graphs with varying degrees of spurious correlations.
> > >
> >
> > A4. Thank you for bringing up this insightful point! We acknowledge that your idea makes great sense, especially for modeling the intervention graph embedding. Discovering causal graph-architecture relationships through graph properties and applying varying levels of intervention based on these properties aligns well with our approach. Since our main focus is still on the overall process of neural architecture search, your suggested method could potentially improve the performance further. We would like to explore and incorporate this idea in future.
> >
> > > Q2. The overall objective function (Eq.(17)) uses a linearly growing $\sigma_p$ corresponding to epoch number. Could you elaborate on why linear growth was chosen over other schedules (e.g., exponential, step-wise)? How does the schedule of  $\sigma_p$ affect the trade-off between causal structure learning and architecture optimization?
> > >
> >
> > A5. Thank you for your detailed question. Actually, we choose linear growth just for simplicity and other methods you mentioned can also be adopted, yet the choice of scheduling method may not significantly impact overall performance. The purpose of adopting this $\sigma_p$ schedule is to enhance training efficiency by dynamically adjusting the focus of training in each epoch—emphasizing the causal-aware part (i.e., identifying suitable causal subgraphs and learning vectors of operators) in the early stages, and prioritizing the performance of the customized super-network in the later stages.
> >
> > We further report both the training loss and validation loss of the two components ($\mathcal{L}\_{causal}$ and $\mathcal{L}\_{pred}$) with and without the dynamic $\sigma_p$ schedule in **`Figure 5`**. This analysis verifies and illustrates the impact of $\sigma_p$ on the trade-off between causal structure learning and architecture optimization. The *newly added detailed analysis is highlighted in blue* in `Appendix E.3` in the **updated [`pdf`](https://openreview.net/pdf?id=58AhfT4Zz1)**.
> >
> > [1] Discovering invariant rationales for graph neural networks, ICLR 22.
> >
> > [2] Learning Invariant Graph Representations for Out-of-Distribution Generalization, NeurIPS 22.

---

### Official Review · Reviewer_n9L7 · 2024-11-04

**Soundness:** 2
**Presentation:** 3
**Contribution:** 2
**Rating:** 6
**Confidence:** 3

**Summary:**

The paper proposes a novel method, Causal-aware Graph Neural Architecture Search (CARNAS), to enhance the generalizability of Graph Neural Network (GNN) architectures under distribution shifts. By discovering stable causal relationships between graph structures and GNN architectures, CARNAS aims to mitigate issues with spurious correlations that often degrade performance across varying distributions. CARNAS introduces three core modules: Disentangled Causal Subgraph Identification, Graph Embedding Intervention, and Invariant Architecture Customization. Experimental results on both synthetic and real-world datasets show significant performance gains, especially in out-of-distribution generalization.

**Strengths:**

1. Well-structured modular approach: The CARNAS framework is thoughtfully organized, with each component clearly contributing to improved generalization under distribution shifts.

2. Robust experimentation: The paper includes extensive experiments across synthetic and real-world datasets, highlighting the robustness of the proposed method.

3. Component-level contribution clarity: Each module’s individual contribution is demonstrated, providing transparency and supporting the effectiveness of the approach.

**Weaknesses:**

1. Clarity in Section 3.3: Given I’m having limited familiarity with Graph NAS, the dynamic graph neural network architecture production and optimization process described in Section 3.3 remains somewhat unclear for me. A visual representation and a more detailed explanation would significantly improve the paper's readability.

2. Causal-Aware Solution's Justification: While the paper presents a causal-aware solution for handling distribution shifts, some aspects require stronger theoretical support to underscore the novelty and significance of the approach:

2.1. Limited Theoretical Support: The causal-aware Graph NAS solution leans heavily on implementation specifics, which limits the theoretical grounding of the method and may impact the perceived novelty.

2.2. Reliability of Causal Subgraph in Latent Space: The representation of causal subgraphs in latent space is an interesting approach; however, it is not entirely clear if the model reliably learns the true causal components or overfits to the training set to optimize the objective.

2.3. Overlap with Prior Work: Section 3.1 closely mirrors aspects of PGExplainer [26], which limits the novelty of this part of the approach.

**Questions:**

1. In Line 113, $N_{tr}$ and $N_{te}$ are not explicitly defined, though their meanings seem clear from the context. Please clarify these terms in the final manuscript.

2. How are the subgraphs in Eq. 6 represented (e.g., soft edge mask or hard crop)? Where and how are they used in subsequent steps?

3. To enhance the clarity of your approach, it would be helpful to visualize the causal and non-causal subgraphs for each dataset used in the case studies.

---

> ### Author Response · Authors · 2024-11-26
>
> We would like to express our sincere appreciation to the reviewer for providing us with detailed comments and suggestions. We have carefully reviewed each comment and offer the following responses:
>
> >W1. Clarity in Sec 3.3: Given I’m having limited familiarity with Graph NAS, the dynamic graph neural network architecture production and optimization process described in Sec 3.3 remains somewhat unclear for me. A visual representation and a more detailed explanation would significantly improve the paper's readability.
>
> A1. Thank you for your valuable feedback. We aim to intuitively illustrate the architecture production and optimization process below, referring to `Figure 1`, which visualizes Section 3.3, along with the **algorithm** in `Appendix B`.
> - Production: Given a graph embedding, such as $\mathbf{H_c}$, our goal is to construct an architecture $A$ (with $K$ layers, each containing $|\mathcal{O}|$ candidate operators). For example, in `Figure 1`, the architecture consists of 2 layers, each with 3 candidate operators. We represent the architecture $A$ as a matrix $\mathbf{A} \in \mathbb{R}^{K \times |\mathcal{O}|}$, where $\mathbf{A}_{k,u} = \alpha_u^k$ is the mixture coefficient of operator $o_u(\cdot)$ in layer $k$, denoting the importance of different operators in different layers. In the figure, this coefficient is visualized as arrows of varying shades in the architecture. The constructed architecture $A$ is the result of combining these coefficients with the specific operators, as shown in Eq. 11. Additionally, we use a prototype vector $\mathbf{op}_u^k$ to represent an operator in a specific layer, allowing us to learn the coefficient $\alpha_u^k$ based on both $\mathbf{op}_u^k$ and $\mathbf{H_c}$.
> - Optimization: Using the production process described above, we customize the optimal architecture $A_c$ for $G_c$ and simulate intervention architectures ${A_v}j$s for intervention graphs. We use $A_c$ as the optimal architecture for the graph instance to output the final label prediction and apply $\mathcal{L}_{arch}$ to ${A_v}_j$s to **regulate the influence of spurious components when constructing the optimal architecture**. This approach identifies and leverages the causal relationships between graphs and architectures to construct the optimal architecture.
>
> >W2. Causal-Aware Solution's Justification: While the paper presents a causal-aware solution for handling distribution shifts, some aspects require stronger theoretical support to underscore the novelty and significance of the approach.
>
> >W2-1. More Theoretical Support.
>
> A2-1. Thank you for your feedback. Although the rebuttal time is limited, we have attempted to provide a detailed theoretical analysis of the problem to further illustrate our method in a more rigorous manner. The added theoretical analysis is highlighted in blue in `Appendix C` in the **updated [`pdf`](https://openreview.net/pdf?id=58AhfT4Zz1)**.
>
> >W2-2. Reliability of Causal Subgraph in Latent Space.
>
> A2-2. Thank you for your comment. Representing causal subgraphs and performing interventions in the latent space allow us to construct architectures based on the intervened latent features. Following your advice, the added theoretical analysis also reflects how our method obtains the causal parts. Additionally, we visualize the edge importance for selecting causal subgraphs in each graph in `Figure 8` and the searched architectures for them in `Figure 7` (`Appendix G`), which better demonstrate the learned causal graph-architecture relationships.
>
> >W2-3. Difference with PGExplainer.
>
> A2-3. Thank you for your comment. While PGExplainer [26] aims to identify explanatory subgraphs based on edge features—a common step in various graph tasks—we would like to clarify the key differences between our method and PGExplainer.
>
> In the **Disentangled Causal Subgraph Identification** module, we introduce the concept of disentangled learning [1,2] to represent distinct latent factors as unique vector representations. This approach allows us to comprehensively disentangle causal latent graph structural features from spurious ones, enabling a more precise identification of causal subgraphs that influence architecture construction. Together with the subsequent modules, our method effectively **disentangles complex graph-architecture relationships**, learning distinct latent factors that guide the customization of optimal architectures from intricate graph structure features.
>
> To summarize, CARNAS **differs in both objective and methodology**: 1. Objective and Motivation: PGExplainer focuses on extracting explanatory subgraphs, while our method aims to disentangle causal features from spurious ones and extract causal subgraphs for architecture optimization. 2. Mechanism: Instead of merely selecting important edges based on graph features, we leverage a disentangled layer to separate causal features from spurious ones, offering a fundamentally different mechanism for subgraph identification.

---

> > ### Author Response · Authors · 2024-11-26
> >
> > > Q1. In Line 113, $N_{tr}$ and $N_{te}$ are not explicitly defined, though their meanings seem clear from the context. Please clarify these terms in the final manuscript.
> > >
> >
> > A3. Thank you for pointing this out. In the updated version, we have clarified that $N_{tr}$ and $N_{te}$ represent the number of graph instances in the training set and testing set, respectively.
> >
> > > Q2. How are the subgraphs in Eq. 6 represented (e.g., soft edge mask or hard crop)? Where and how are they used in subsequent steps?
> > >
> >
> > A4. Thank you for your question. In Eq. 6, the subgraphs are extracted using a hard crop on edges to identify the causal subgraph $G_c$, while simultaneously separating out spurious subgraphs ${G_s}_j,~j \in [1, N_s]$.
> >
> > Specifically, in our method: The causal subgraph $G_c$, extracted from the *Disentangled Causal Subgraph Identification module*, serves as the carrier of information causally relevant to the optimal architecture. Using the *Graph Embedding Intervention module*, spurious subgraphs ${G_s}_j$s are employed to generate intervention graphs ${G_v}_j$s (with graph representations $\mathbf{H_v}_j$ ) by intervening on $G_c$. In the *Invariant Architecture Customization module*, the optimal architecture $A_c$ is customized with $G_c$ and simulated intervention architectures ${A_v}_j$s are customized with intervention graphs. The loss function $\mathcal{L}\_{arch}$ for ${A_v}_j$s can **regulate the influence of spurious parts on the construction of the optimal architecture**. Through this process, both causal and spurious subgraphs are integral to identifying and leveraging causal graph-architecture relationships for constructing the optimal architecture.
> >
> > > Q3. To enhance the clarity of your approach, it would be helpful to visualize the causal and non-causal subgraphs for each dataset used in the case studies.
> > >
> >
> > A5. Thank you for your suggestion! Following your advice, we have visualized the causal subgraphs for each dataset used in the case study in `Appendix G, Figure 8`. These visualizations illustrate the learned graph-architecture relationships effectively. Please refer to the **updated [`pdf`](https://openreview.net/pdf?id=58AhfT4Zz1)** for more details.
> >
> >
> > [1]Representation learning: A review and new perspectives.
> >
> > [2]Disentangled representation learning.

---

### Author Response · Authors · 2024-12-04
**Summary**

Dear reviewers,

We would like to summarize the rebuttal and revised paper as below:

1. We provide a **detailed theoretical analysis** of the problem to further illustrate our method in a more rigorous manner. The added theoretical analysis is highlighted in blue in `Appendix C` in the **updated [`pdf`](https://openreview.net/pdf?id=58AhfT4Zz1)**.
2. We have **visualized the causal subgraphs** for each dataset used in the case study in `Appendix G, Figure 8`, in the **updated [`pdf`](https://openreview.net/pdf?id=58AhfT4Zz1)**. These visualizations illustrate the learned graph-architecture relationships effectively.
3. We further report both the training loss and validation loss of the two components ($\mathcal{L}\_{causal}$ and $\mathcal{L}\_{pred}$) with and without the dynamic schedule in **`Figure 5`**. This analysis verifies and illustrates the impact of the dynamic schedule on the trade-off between causal structure learning and architecture optimization. The *newly added detailed analysis is highlighted in blue* in `Appendix E.3` in the **updated [`pdf`](https://openreview.net/pdf?id=58AhfT4Zz1)**.
4. We have included new results on the **GOODSST2 dataset**, including **11 OOD-GNN baselines**, comparing both performance, time and memory costs, in `Appendix F` in the revised [**`pdf`**](https://openreview.net/pdf?id=58AhfT4Zz1). Results indicate that **CARNAS achieves the best performance while maintaining competitive time and memory costs compared to non-NAS-based (fixed-network) graph OOD methods.** The efficiency of the architecture search process stems from CARNAS's ability to **simultaneously search for the architecture and optimize its parameters, minimizing additional computational overhead**. This demonstrates the practical viability and efficiency of CARNAS, even for larger-scale graph datasets.

We sincerely thank all the reviewers for dedicating your valuable time and effort to evaluate our work. We truly hope the responses have clarified your concerns and contributed to a better understanding of our research.

---

### Meta-Review · Area_Chair_ajR6 · 2024-12-07

**Metareview:**

This paper proposes a causal-aware graph NAS framework to address the challenge of distribution shifts. By leveraging causal relationships between graph structures and architectures, the method aims to mitigate reliance on spurious correlations and enhance out-of-distribution generalization.  Extensive experiments on synthetic and real-world datasets demonstrate its superior performance over existing methods in OOD settings.

The strengths of the work include the new scenario that combines Graph NAS and OOD generalization and the evaluation that shows the generally better performance.  The weaknesses include the presentation that causes confusions on two reviewers, the incremental contribution in techniques that leverage methods directly from existing OOD graph learning works (e.g., disentangled learning) and from NAS techniques (e.g., DARTS), and some concerns on the complexity.

Based on the strengths and weaknesses, I recommend rejection for this submission in its current form, given the high bar of ICLR. That said, I agree that the idea of combining causality with Graph NAS is innovative and the experimental results are promising. Addressing the concerns and having OOD techniques more relevant to the NAS scenario could lead to a strong contribution.

**Additional Comments On Reviewer Discussion:**

Two reviewers mentioned the concern of the complexity of the method. The authors did more experiments to show that the method is not substantially slower than other models with fixed architecture. However, the authors did not try the large graph cases. The concerns on the technical novelty were also raised by two reviewers. The authors provided some high-level explanations. If the authors could better clarify that with more rigorous discussion, via formulas, theorems, etc. It might be more helpful for others to appreciate it.

---

### Decision · Program_Chairs · 2025-01-22

Reject